# *Mettl3*-dependent m⁶A modification attenuates the brain stress response in *Drosophila*

Alexandra E. Perlegos [1], Emily J. Shields[2,3,4], Hui Shen[5], Kathy Fange Liu [5,6] & Nancy M. Bonini [1,2] ✉

$N^6$-methyladenosine (m⁶A), the most prevalent internal modification on eukaryotic mRNA, plays an essential role in various stress responses. The brain is uniquely vulnerable to cellular stress, thus defining how m⁶A sculpts the brain's susceptibility may provide insight to brain aging and disease-related stress. Here we investigate the impact of m⁶A mRNA methylation in the adult *Drosophila* brain with stress. We show that m⁶A is enriched in the adult brain and increases with heat stress. Through m⁶A-immunoprecipitation sequencing, we show 5′UTR *Mettl3*-dependent m⁶A is enriched in transcripts of neuronal processes and signaling pathways that increase upon stress. *Mettl3* knockdown results in increased levels of m⁶A targets and confers resilience to stress. We find loss of *Mettl3* results in decreased levels of nuclear m⁶A reader *Ythdc1*, and knockdown of *Ythdc1* also leads to stress resilience. Overall, our data suggest that m⁶A modification in *Drosophila* dampens the brain's biological response to stress.

The cellular stress response is among the most ancient pathways, critical for all cells to combat insults. Despite this, prolonged exposure to cellular stress is detrimental, in particular for the maintenance of brain integrity and function[1–4]. During cellular stress, transcription and protein synthesis is globally repressed to reduce cell burden[5]. Concurrently, a subset of mRNAs undergoes selective translation to prioritize synthesizing proteins vital to stress recovery and disease suppression[6]. Neurons employ altered mechanisms to adapt and survive stress. These include delayed activation of HSF-1 in motor neurons[7,8] and activation of pathways that promote axon regeneration, neuronal apoptosis, and brain longevity[9–11]. As neurons are postmitotic and cannot dilute potentially harmful species through cell division, they must adapt conserved pathways such as heat shock (HS) chaperones, mTOR signaling, and Hippo signaling to selectively maintain synaptic and dendritic homeostasis[12–14]. Given the striking susceptibility of the brain to neurodegenerative disease, understanding the

effectors of the adult brain stress response in vivo may be critical to understanding brain disease and healthful brain aging.

Epigenetic modification of DNA and histone proteins are crucial in neuronal responses to cellular stress and disease[15–17]. However, recent techniques have provided increasing evidence that RNA modifications confer a critical layer of regulation[18–20]. m⁶A is the most abundant internal modification of mRNA in eukaryotic cells and is deposited co-transcriptionally by a methyltransferase, *Mettl3*, with the help of additional complex components[21]. m⁶A-modified transcripts are recognized by selective reader proteins that dictate downstream processing, such as splicing, decay, promotion and inhibition of translation[22–25]. In vitro work in mammalian cells has shown that m⁶A is dynamically regulated during stress, including upon UV DNA damage and heat-shock[26–29]. m⁶A-marked mRNAs are enriched in stress granules and RNA-protein complexes that form during cellular stress, and m⁶A YTH reader proteins are fundamental for stress granule

[1]Neuroscience Graduate Group, University of Pennsylvania, Philadelphia, PA 19104, USA. [2]Department of Biology, University of Pennsylvania, Philadelphia, PA 19104, USA. [3]Epigenetics Institute and Department of Cell and Developmental Biology, University of Pennsylvania Perelman School of Medicine, Philadelphia, PA, USA. [4]Department of Urology and Institute of Neuropathology, Medical Center-University of Freiburg, Faculty of Medicine, University of Freiburg, Freiburg, Germany. [5]Department of Biochemistry and Biophysics, Perelman School of Medicine, University of Pennsylvania, Philadelphia, PA 19104, USA. [6]Graduate Group in Biochemistry and Molecular Biophysics, Perelman School of Medicine, University of Pennsylvania, Philadelphia, PA 19104, USA. ✉e-mail: nbonini@sas.upenn.edu

formation[30–33]. In vivo studies in mice show that neuronal m⁶A is essential for axon regeneration post-injury and stress-related psychiatric disorders[34,35]. Although m⁶A has been widely shown to influence RNA stability and translation, its role is still being defined, with varying studies highlighting alternative mechanisms[24,26,27,36].

*Drosophila* is a powerful model system with many processes conserved with mammals, including brain development and function. In *Drosophila*, m⁶A regulates sex-determination by modulating mRNA splicing during development[37]. The m⁶A methyltransferase complex components are conserved and selectively enriched in the *Drosophila* central nervous system[38]. Recent work also implicates the *Drosophila* cytoplasmic reader protein Ythdf in the regulation of learning and memory, and nervous system development through interactions with fragile X syndrome protein Fmr1[39,40].

Here we investigated m⁶A regulation of the heat stress response of the adult fly brain. We find that m⁶A is enriched in the 5′UTR, and 5′UTR modification is regulated by the methyltransferase *Mettl3*. *Mettl3* knockdown increases protein and RNA levels of its targets and confers unexpected stress resilience. This work highlights that *Mettl3*-dependent m⁶A modification normally serves to dampen the brain's acute heat stress response by selective regulation of RNA levels and translation of critical signaling and cellular stress transcripts.

## Results

### m⁶A levels are enriched in the brain and are modulated with heat stress

The stress response is critical to the brain's ability to endure the challenges of age and disease, and m⁶A modulation of transcripts is associated with a range of stress responses in mammalian cells. Molecular chaperones crucial to the heat shock response are instrumental in understanding neurological disease, and preconditioning heat stress has been noted to suppress disease toxicity and increase longevity[3,41,42]. Therefore, we examined if there were changes in m⁶A upon the brain heat shock (HS) response in vivo. We first profiled global m⁶A with heat stress in head tissue, which is brain-enriched. Intriguingly, the levels of m⁶A were increased on polyadenylated RNA (polyA + RNA) after HS. The m⁶A signal was robust after 10, 30, and 60 min of HS and returned to baseline by 24 h of recovery, indicating that m⁶A levels were dynamically changed with HS (Fig. 1a, b).

To investigate this response further, we focused on the 30 min HS timepoint, as it showed the most significant increase in m⁶A levels. Animals lacking eye tissue (a large fraction of the head tissue) showed a similar robust increase in m⁶A levels with HS (Supplementary Fig. 1a), suggesting that the signal was largely from brain tissue and not the eyes. By liquid chromatography-tandem mass spectrometry (LC-MS/MS), we confirmed an increase in m⁶A with HS from dissected brains (Fig. 1c). The HS-related change in m⁶A levels was not detected in polyA+ RNA isolated from whole fly tissue (Fig. 1d). Using LC-MS/MS, m⁶A levels were 2.2 × higher in the brain compared to the whole fly (Fig. 1e), consistent with previous studies showing that fly m⁶A enzymes are enriched in the central nervous system[38]. These data indicate that m⁶A is enriched in *Drosophila* brain tissue, and changes in m⁶A with stress are primarily in the brain.

To define the extent to which the m⁶A change was due to methyltransferase *Mettl3* activity, we knocked down *Mettl3* by RNAi (Supplementary Fig. 1b–d). This reduced m⁶A levels on polyA⁺ RNA and reduced the increase seen with HS (Fig. 1b), indicating that the dynamic change of m⁶A on polyA+ RNA with stress is largely dependent on *Mettl3* gene function. Curiously, when we examined total RNA rather than polyA+ RNA, we found that m⁶A levels in total RNA were decreased upon HS, with levels returning to normal after 24 h (Supplementary Fig. 1e). This decrease was not impacted by *Mettl3* RNAi, indicating that *Mettl3* methyltransferase activity does not affect total RNA (consisting mostly of rRNA) m⁶A methylation (Supplementary Fig. 1f). Overall, these findings signify that m⁶A increases on polyA+

RNA in the brain upon HS in a manner dependent on *Mettl3* gene function.

### The brain has a distinct stress response

Given these data showing that the brain has a unique m⁶A response to HS, we investigated mRNA and protein changes of selected molecular chaperones known to increase with stress. Whole fly and whole head tissue undergo a typical *Drosophila* HS response[41], with rapid upregulation of Hsp70 protein by 30 min of HS that persists through a 6 h recovery (Supplementary Fig. 2a). However, the brain (dissected from the head capsule and eyes) showed a delayed upregulation of Hsp70 protein compared to whole fly and whole head (Supplementary Fig. 2a). Additionally, the protein levels of Hsp70 upon heat stress were lower in the brain relative to other tissues, specifically whole heads, and outer head capsules minus brain (Supplementary Fig. 2b, c). We investigated two additional fly chaperones, *DnaJ-1* (homolog of mammalian HSP40; Supplementary Fig. 2d) and *stv* (homolog of mammalian BAG3; Supplementary Fig. 2e), as these two proteins function to stimulate heat shock protein chaperone function[43,44]. Again, surprisingly, their protein-level upregulation was markedly dampened in the brain, with no significant increase in protein by Western immunoblot upon HS. By contrast, there was a similar strong transcriptional upregulation of all of these chaperones in all tissues examined (Supplementary Fig. 2f–h). These findings indicate a striking disconnect between RNA and protein levels of key chaperones upon HS in the brain and indicate that the brain has a dampened response to stress at the protein level compared to other tissues.

### *Mettl3* knockdown animals are resilient to heat stress

Based on the enrichment of m⁶A in the brain, m⁶A modulation upon stress, and the diminished chaperone protein response of the brain, we reasoned that m⁶A reduction may impact stress resilience. We performed an animal HS survival assay by subjecting control and *Mettl3* RNAi male flies to severe HS (1.5 h at 38.5 °C), then scoring fly survival at 24 h (Fig. 2a). Approximately 50–60% of control animals survived 24 h after HS (Fig. 2b). Upon reduction of *Mettl3*, however, the animals were notably more resilient, showing a 10–20% increase in survival compared to controls treated in parallel (Fig. 2b). Additionally, knockdown of *Mettl3*'s heterodimeric partner *Mettl14* also conferred stress resilience (Supplementary Fig. 3a). By contrast, upregulation of *Mettl3* selectively in neurons dramatically impaired HS survival, with nearly none of the animals surviving 24 h post-HS (Fig. 2c). The *Mettl3* ΔCat (catalytic domain deletion[38]) mutant in trans to a *Mettl3* deficiency was also stress resilient (Fig. 2d and Supplementary Fig. 3b). We examined protein levels of Mettl3 upon HS and found no significant changes (Supplementary Fig. 3c). The resilience of the *Mettl3* ΔCat and *Mettl14* knockdown animals indicates this HS pathway is dependent on *Mettl3* m⁶A methyltransferase activity. Overall, these data indicate that m⁶A modification in the brain is associated with reduced HS survival.

### m⁶A is enriched in the 5′UTR of *Drosophila* transcripts

Given the change in m⁶A levels in polyA⁺ RNA and resilience of *Mettl3* knockdown animals upon HS, we sought to define which transcripts are marked with m⁶A upon HS in vivo. m⁶A-immunoprecipitation (IP) sequencing was performed from polyA+ RNA extracted from control and *Mettl3* knockdown heads, in basal and 30 min HS conditions (Supplementary Fig. 4a). Using brains alone was challenging for m⁶A-IP sequencing due to low RNA yield, and consequently we used head tissue, given our data showing m⁶A in the head is associated with brain tissue (see Fig. 1b). To confirm robustness of the IP and targets, m⁶A-modified RNA was assessed using m⁶A antibodies produced by two different sources: New England Biolabs (NEB) and Synaptic Systems (SYS). The data were highly consistent between antibodies, with high overlap (80%) of m⁶A-enriched targets (Supplementary Fig. 4b).

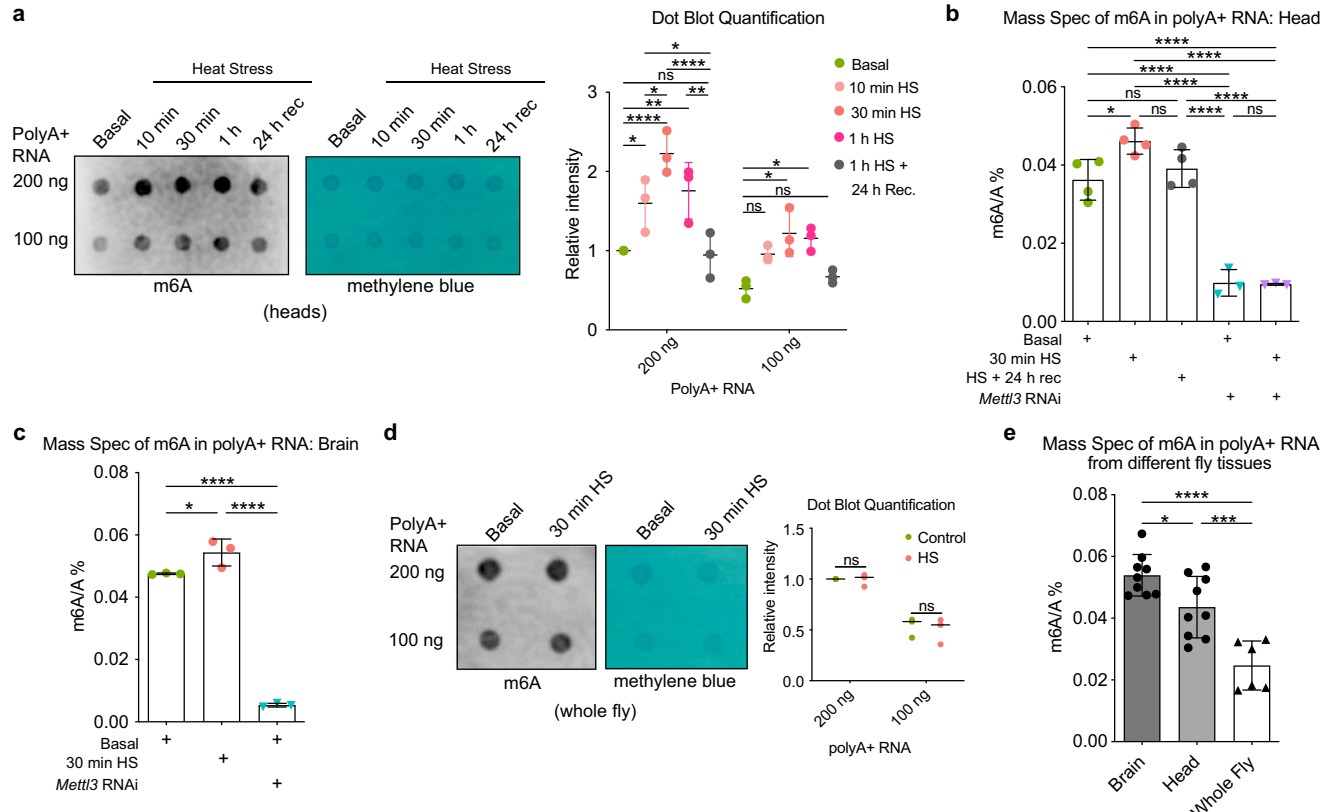

**Fig. 1 | m⁶A levels are enriched in the brain and are modulated with heat stress.** **a** Dot blot assay of polyA+ RNA m⁶A levels in fly heads, $w^{1118}$ (BL5905), was extracted immediately after HS at 38.5 °C for 10 min, 30 min, 1 h, and HS 1 and 24 h recovery. Quantification of m⁶A levels relative to basal (intensity = 1) and normalized to methylene blue. 1 ul dots of 200 and 100 ng polyA+ from the same replicate are blotted onto the membrane. $n = 3$ biological replicates, 200 heads per sample condition and replicate. Data presented as mean ± SD, $*p < 0.05$, $**p < 0.01$, $***p < 0.001$, $***p < 0.0001$, two-way ANOVA with Sidak's test. In **a** from left to right comparisons: $p = 0.0459$, $p < 0.0001$, $p = 0.0066$, $p = 0.0314$, $p = 0.0034$, $p < 0.0001$, $p = 0.0314$, $p = 0.0138$, $p = 0.0296$, ns not significant. **b** LC-MS/MS analysis of m⁶A/A% levels from head polyA+ RNA in basal and HS 30 min, in DaGal4>$Mettl3$ RNAi vs DaGal4 > mCherry RNAi conditions, $n = 3$ biological replicates for $Mettl3$ RNAi conditions, $n = 4$ for control animal conditions, 30 heads per replicate. Data presented as mean ± SD, $*p < 0.05$, $****p < 0.0001$, one-way ANOVA with Tukey's test. In **b** from left to right comparisons: $p = 0.0266$, $p < 0.0001$, ns not significant, $p < 0.0001$, $p < 0.0001$, $p < 0.0001$, $p < 0.0001$, $p < 0.0001$. **c** LC-MS/MS analysis of m⁶A /A levels in polyA+ RNA from the brain in basal, 30 min HS, and $Mettl3$ RNAi in basal conditions. $n = 3$ biological replicates, 40 brains per replicate, DaGal4>$Mettl3$ RNAi vs mCherry RNAi. Data presented as mean ± SD, $*p < 0.05$, $***p < 0.01$, $****p < 0.0001$, one-way ANOVA with Tukey's test. In **c** from left to right comparisons: $p = 0.0377$, $p < 0.0001$, $p < 0.0001$. **d** Dot blot of the whole fly $w^{1118}$ (BL5905) polyA+ RNA m⁶A levels with HS 30 min. Quantification of m⁶A levels normalized to control and methylene blue. $n = 3$ biological replicates, ten whole flies per sample condition and replicate. Data presented as mean, two-way ANOVA with Sikak's test, ns not significant. **e** LC-MS/MS analysis of m⁶A /A levels in polyA+ RNA from brain, head, and whole fly of male flies $w^{1118}$ (BL5905). For brains $n = 9$ biological replicates with 40 brains per replicate, for heads $n = 9$ biological replicates with 30 heads per replicate, for whole flies $n = 6$ biological replicates with eight whole flies replicate. Data presented as mean ± SD, $***p < 0.001$, $****p < 0.0001$, one-way ANOVA with Tukey's test. $p = 0.0404$, $p < 0.0001$, $p = 0.0009$. Additional details for fly genotypes for each figure are provided in Supplementary Data 7. Source data and statistical analysis are provided as a Source Data file.

We found that m⁶A, which is dependent on $Mettl3$, was greatly enriched in the 5′UTR, with less density along coding sequences (CDS) and 3′UTR (Fig. 3a). The enrichment of m⁶A in the 5′UTR may be unique to *Drosophila* and is consistent with other recent works[39,40], in contrast to mammalian m⁶A that is mostly in the 3′UTR with modest 5′UTR increase during stress[34,45]. De novo motif analysis highlighted similar m⁶A motifs from both antibody m⁶A-IPs (Supplementary Fig. 4c and Supplementary Data 1) that were consistent with motifs defined from other *Drosophila* m⁶A-IP seq studies[38,39,46]. We used the SYS antibody for further analyses for ease of comparison to other sequencing studies using SYS. We used RADAR[47] differential methylation analysis to call $Mettl3$-dependent m⁶A peak changes between control and $Mettl3$ knockdown conditions; these targets will be referred to as m⁶A genes from here on (Supplementary Data 2). Our global analysis of m⁶A marked transcripts showed a slight enrichment of 5′UTR m⁶A upon heat stress, consistent with our LC-MS/MS and dot blot analyses (Fig. 3a, b and Supplementary Fig. 4d).

Knockdown of $Mettl3$ showed 94% of $Mettl3$-dependent m⁶A was located in the 5′UTR (Fig. 3b–d). The m⁶A targets affected by $Mettl3$

knockdown, defined here by m⁶A-IP-seq, showed considerable overlap (79%) with *Drosophila* head m⁶A miCLIP-seq targets[39] (Supplementary Fig. 4e). There was less overlap (47%) with S2R + cell culture m⁶A miCLIP-seq targets[40] (Supplementary Fig. 4f). These data underscore the enrichment of m⁶A in the 5′UTR in *Drosophila* transcripts and highlight that the identity of m⁶A-marked genes depends on the genes expressed in vivo vs cell culture conditions. Overall, our sequencing analysis showed consistency between replicates (Supplementary Fig. 4g).

## m⁶A transcripts are enriched in neuronal and signaling pathways

To define and understand the types of genes regulated by m⁶A modification, we compared genes with m⁶A (a majority of which are in the 5′ UTR), to all other genes expressed in the brain. This comparison indicated that only 14.1% of all genes expressed in the brain are marked by m⁶A.

Unbiased GO term and KEGG pathway analyses highlighted that m⁶A transcripts were distinct from non-modified brain genes. Non-

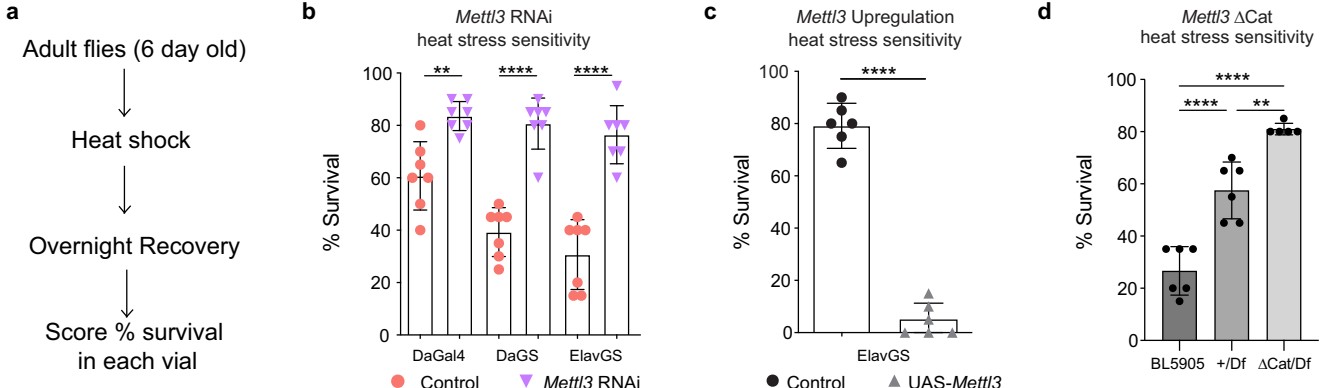

**Fig. 2 | Mettl3 knockdown animals are resilient to heat stress. a** Schematic of HS stress assay. Flies are 6-day-old. Flies are put onto RU486 food if needed to turn on the expression of the RNAi transgene for 6 days. Flies are HS at 38.5 °C and allowed to recover overnight in normal food or RU486 food at 25 °C. Percent survival in each vial of 20 animals are scored after 24 h. **b** Ubiquitous knockdown of *Mettl3* (DaGal4, DaGS), Neuron-specific knockdown (ElavGS), and corresponding controls (mCherry RNAi) were HS for 1.5 h at 38.5 °C and scored for survival after 24 h recovery. *n* = 7 biological replicates, each data point represents percent survival in a vial of 20 flies per replicate. Data presented as mean ± SD, **\*\*p < 0.01, \*\*\*\*p < 0.0001, Student's two-tailed *t*-test. *p* = 0.0011, *p* < 0.0001, *p* < 0.0001. **c** Neuron-specific (ElavGS) upregulation of *Mettl3* and control were HS for 1 h at 38.5 °C and scored for

survival after 24 h recovery. *UAS-Mettl3* flies were severely stress-sensitive, and HS time was reduced to 1 h to have at least 5–10% of the flies surviving. Each data point represents percent survival in vials of 20 flies per replicate. Data presented as mean ± SD, \*\*\*\*p < 0.0001, Student's two-tailed *t*-test, *p* < 0.0001. **d** Catalytic *Mettl3* Mutant (ΔCat /Df) vs *Mettl3* deficiency (Df/+) vs *w^1118* (BL5905) (+/+) were HS for 1.5 h at 38.5 °C and scored for survival after 24 h recovery. Each data point represents percent survival in vials of 20 flies per replicate. Data presented as mean ± SD, \*\*p < 0.01, \*\*\*\*p < 0.0001, One-way ANOVA with Tukey's test. *p* < 0.0001, *p* < 0.0001, *p* = 0.0014. Source data and statistical analysis are provided as a Source Data file.

modified brain genes were enriched for "metabolic process" and "oxidative phosphorylation" terms (Supplementary Fig. 5a, b and Supplementary Data 3), while genes with m⁶A were enriched in GO terms associated with "neurodevelopment" and "neurogenesis" (Fig. 3e, left). KEGG pathway analysis indicated that m⁶A genes were enriched for critical signaling pathways, including MAPK, WNT, TGF-beta, and Notch, among others (Fig. 3e, right). Intriguingly, neurons often employ these signaling pathways to deal with stressful situations[10,11]. We visualized m⁶A enrichment across the gene body by calculating the m⁶A-IP reads divided by the background input reads for genes annotated with select KEGG pathway terms (Fig. 3f). These data emphasized that m⁶A genes involved in signaling pathways are enriched for m⁶A in the 5′UTR, consistent with the global analysis.

To further investigate the impact of m⁶A modification on brain gene expression, we examined protein levels of m⁶A modified genes with available antibodies. We chose three genes: *fl(2)d* (WTAP, a component of the m⁶A writer complex), *futsch* (MAP1A/B, a microtubule-associated protein that regulates axonal growth), and *draper* (a receptor that is involved in response to stimuli and phagocytosis by glia) (Supplementary Fig. 5c). Protein levels of all three genes increased in brain tissue upon *Mettl3* knockdown compared to controls in basal and in HS conditions (Supplementary Fig. 5d–f). Brain transcript and protein levels of the non-modified gene *Hsf*, the major transcription factor for HS stress, showed no significant difference with *Mettl3* RNAi (Supplementary Fig. 5g). These data suggest that depletion of *Mettl3* is associated with increased protein levels of m⁶A-target genes in the brain.

Comparing the transcriptome of epithelial S2 cells, which previous studies have focused on refs. 38, 48, to brain tissue, we found that m⁶A-marked genes are more highly expressed in the brain (Supplementary Fig. 5h and Supplementary Data 4). These data underscore that m⁶A genes are enriched in brain-specific pathways, indicated also from GO and KEGG analysis, and highlight the importance of defining the in vivo response.

## m⁶A transcripts are increased with HS and upon *Mettl3* knockdown

To further investigate the stress resilience of *Mettl3* knockdown animals and increased m⁶A observed with HS, we performed brain tissue

RNA-seq upon *Mettl3* knockdown and with HS (Supplementary Fig. 6a–d and Supplementary Data 5). We analyzed the global transcriptional response with an RNA-seq time-course on dissected brain tissue from animals exposed to 30 min HS and allowed to recover for 6 and 24 h. The brain showed a strong transcriptional response to HS at 30 min, with similar numbers of genes up- and downregulated (Supplementary Data 5). After 6 and 24 h of recovery, the transcriptional response to the initial heat shock was greatly attenuated, although 16–20% of the genes did not return completely to baseline levels.

We examined m⁶A targets to probe the transcriptional response of this gene set in the brain. Surprisingly, m⁶A genes were preferentially upregulated with HS (1016 genes up and 227 down), compared to all other genes expressed in the brain, which did not show a skew towards increased transcription levels (Fig. 4a and Supplementary Fig. 6e). Examining the transcriptional dynamics of specific groups of m⁶A genes (see Fig. 3e), neurogenesis and MAPK signaling genes containing m⁶A were more upregulated with HS (Fig. 4b). Globally, of the 3193 genes upregulated with HS in the brain, 31.8% had m⁶A modification (Fig. 4c). The skewed increase in expression upon HS was specific for m⁶A gene sets, as no similar trend was observed when considering non-modified gene sets (Fig. 4d). The m⁶A-tagged genes significantly upregulated with HS (47.9% of all m⁶A genes) could also account, in part, for the increased m⁶A levels detected by dot blot and mass spectrometry with HS (see Fig. 1).

In *Mettl3* RNAi brains at basal conditions, m⁶A genes skewed towards upregulated (78.3% of differentially expressed m⁶A genes were up with *Mettl3* RNAi (Fig. 4e–h and Supplementary Fig. 6f)). This upregulation is also seen in m⁶A gene sets enriched for neurogenesis and MAPK signaling pathways (Fig. 4f–h). These data indicate that m⁶A transcripts are likely to be increased upon both HS and *Mettl3* knockdown in the brain. The increased basal levels of these genes may allow for preconditioning to stress in *Mettl3* knockdown animals, thus conferring the stress resilience observed (see Fig. 2).

## Mettl3 knockdown increases HS chaperone protein levels and decreases their rate of RNA decay

To understand mechanistically how m⁶A influences the brain HS response, we analyzed differential m⁶A methylation in basal vs HS

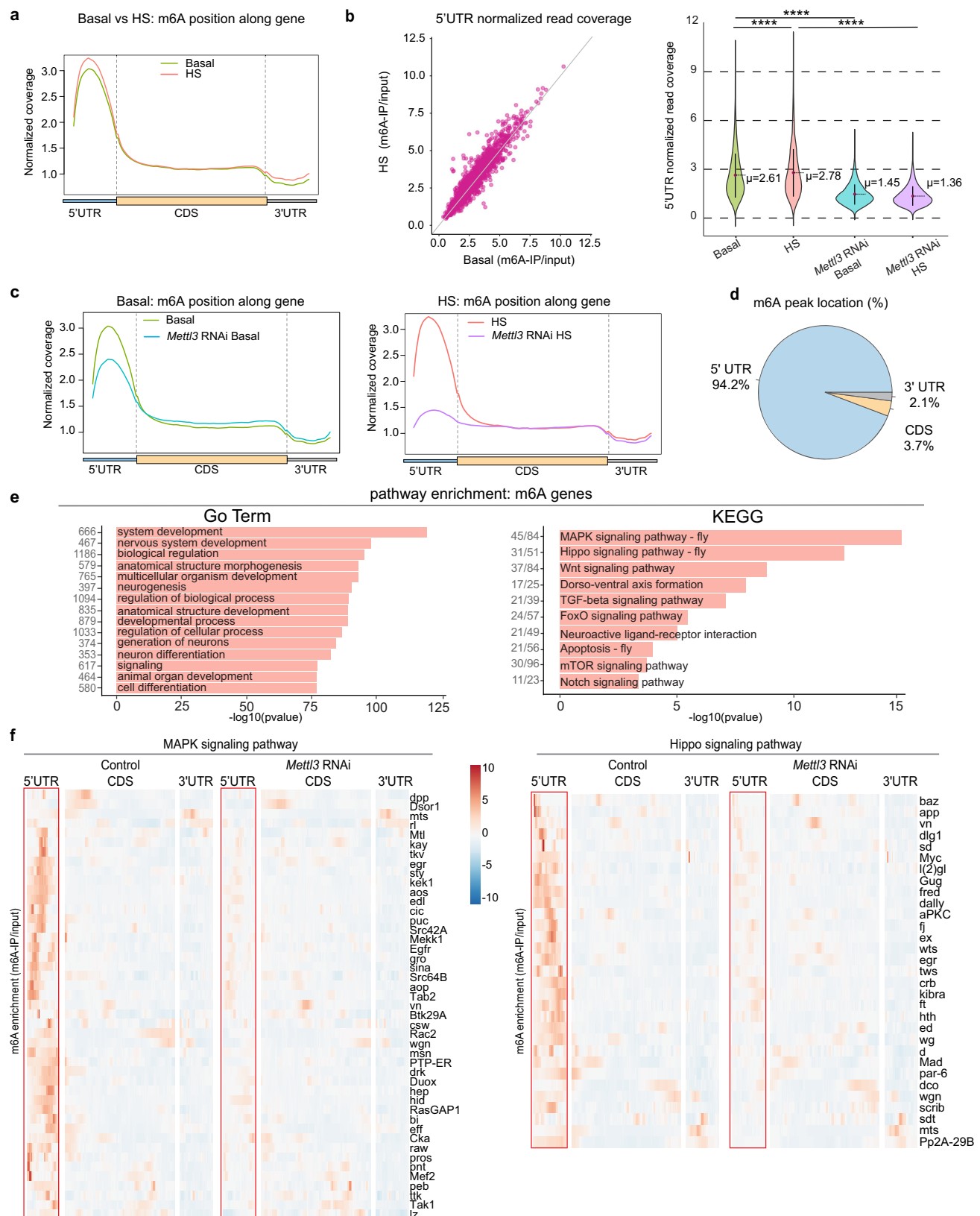

conditions. Although we saw a global shift toward increased m⁶A methylation upon HS (Fig. 3a, b), few genes exhibited statistically significant thresholds of differential methylation upon stress (Supplementary Data 2), consistent with previous m⁶A stress studies in mammals[34,35].

To further understand the impact of m⁶A on HS chaperones, we utilized the FlyBase[49] curated list of HS chaperones, plus 20 of the most significantly differentially expressed genes from our brain HS RNA-seq analysis (Supplementary Data 5 and Supplementary Data 6). We were surprised to find only a few HS chaperones with m⁶A upon HS (red

**Fig. 3 | m⁶A transcripts are enriched in neuronal and signaling pathways.**
**a** Normalized read coverage plot of m⁶A-IP/input on polyA+ transcripts in the 5′ UTR, CDS, and 3′UTR of m⁶A genes (*Mettl3*-dependent m⁶A). m⁶A-IP sequencing in basal and HS conditions from control fly (DaGal4 > mCherry RNAi) heads.
**b** Scatterplot of normalized read coverage of m⁶A-IP/input from the 5′UTR of m⁶A genes in basal versus HS conditions. *n* = 2 biological replicates for each condition, 200 fly heads per replicate per condition. Violin plot showing indicating mean 5′ UTR m⁶A coverage in each condition and statistical significance between each group. Data presented as mean ± SD, ****p < 0.0001, Paired *t*-test and two-tailed. *p* < 0.0001, *p* < 0.0001, *p* < 0.0001. **c** Normalized read coverage plot of m⁶A-IP/ input in basal and HS conditions from DaGal4>*Mettl3* RNAi vs DaGal4>mCherry RNAi samples show a loss of m⁶A primarily in the 5′UTR of transcripts upon

knockdown of *Mettl3*. **d** Percent location of *Mettl3*-dependent m⁶A. *Mettl3*-dependent m⁶A peaks are enriched in the 5′UTR in *Drosophila*. **e** GO term enrichment and KEGG pathway enrichment of m⁶A genes. The −log10(*p* val) enrichment of genes in each category is shown, as well as the number of genes that fall into each category (left). Full GO term, KEGG term, and *p* value lists are shown in Supplementary Data 3. **f** Heat map of m⁶A enrichment on KEGG Pathway: MAPK signaling and hippo signaling genes with *Mettl3*-dependent m⁶A. Shown is mCherry RNAi (control) and *Mettl3* RNAi m⁶A enrichment in basal conditions. m⁶A enrichment is shown as a log (m⁶A-IP divided by the input control). Heat map displays *z*-score values, which is scaled by row; each gene is relative to itself across all six boxes. Segmented into 5′ UTR, CDS, and 3′UTR. Source data and statistical analysis are provided as a Source Data file.

asterisks, Fig. 5a and Supplementary Fig. 7a–c). Perplexingly, the inducible *Hsp70* chaperone genes did not show increased 5′UTR m⁶A upon HS, as in mammalian cell HS studies[26]. Closer examination, however, revealed that several *Hsp70/Hsp68* isoforms showed m⁶A enrichment in the 5′UTR at baseline that was lost with HS and with *Mettl3* RNAi (Fig. 5b and Supplementary Fig. 7a–c). These findings are consistent with both m⁶A-IP antibodies (Supplementary Fig. 7b). We also observed a significant elevation of *Hsp70* transcript levels in *Mettl3* RNAi brains at basal conditions (Supplementary Fig. 7d and Supplementary Data 5). Although *Hsp70* did not pass our conservative thresholds for genes with significant 5′UTR m⁶A (peak height was below the RADAR beta-cutoff threshold of 0.5; the *Hsp68* peak was above this threshold), these observations suggest that 5′UTR m⁶A modulation of *Hsp70/Hsp68* occurs at baseline and is lost upon HS in the *Drosophila* brain.

We examined two additional chaperones critical for Hsp70 function that displayed significant 5′UTR m⁶A. *DnaJ-1* and *stv* were transcriptionally upregulated with HS (Supplementary Data 5) and, unlike *Hsp70*, had significant 5′UTR m⁶A in both basal and HS conditions (Fig. 5c). At baseline, the steady-state levels of *DnaJ-1* and *stv* transcripts were increased in *Mettl3* RNAi brains compared to controls by real-time qPCR (RT-qPCR) (Supplementary Fig. 7e and Supplementary Data 7). Interestingly, differential methylation RADAR analysis indicated *DnaJ-1* and *stv* have significantly decreased 5′UTR m⁶A methylation upon HS (Supplementary Data 2).

To assess the protein level of these chaperones, we dissected brains at baseline, 30 min HS, and recovery at 6 and 24 h post-HS in control and *Mettl3* RNAi conditions (Fig. 5d). Hsp70 showed a significant increase in *Mettl3* RNAi brains with HS and 6 h of recovery (Fig. 5d and Supplementary Fig. 7f). Due to the very low levels of Hsp70 protein at basal conditions, we further examined Hsp70 at baseline on separate blots. We observed a significant increase in *Mettl3* RNAi brains, indicating flies deficient for *Mettl3* have preconditioned or higher levels of the chaperone protein in basal conditions (Supplementary Fig. 7g). DnaJ-1 and stv protein levels were also increased in *Mettl3* RNAi brains at baseline, HS, and with recovery (Fig. 5d). By contrast, upregulation of *Mettl3* led to decreased protein levels of DnaJ-1, stv, and Hsp70 in brains after HS (Supplementary Fig. 7h, i). These data indicate that *Mettl3* function affects the protein levels of these m⁶A marked HS chaperones: knockdown of *Mettl3* leads to upregulation, whereas increased *Mettl3* leads to downregulation.

We hypothesized that altered RNA decay may contribute to the protein level changes of DnaJ-1 and stv in the brain, given the impact of m⁶A on RNA stability in mammalian systems[23,25,50]. To assess RNA decay, we performed an ex vivo brain assay using actinomycin D for transcription inhibition (Supplementary Fig. 8). In the presence of actinomycin D, *Mettl3* RNAi brains showed attenuated decay of *DnaJ-1* and *stv*, while the non-modified transcript RpL32 showed no significant difference (Fig. 5e, f). These data suggest that 5′UTR m⁶A modification may influence RNA transcript levels by decreasing the stability of modified transcripts in the brain. Overall, these key HS chaperones are

examples of m⁶A modified transcripts that increase with HS and in *Mettl3* knockdown brains.

## *Mettl3* attenuates protein translation in the brain

Previous studies in cultured S2R+ cells observed a decrease in puromycin incorporation into nascent proteins upon *Mettl3* knockdown[39], in contrast to our brain data that suggest *Mettl3* loss causes an increased transcript and protein levels of m⁶A modified genes and stress chaperones (see Figs. 3–5). Therefore, we examined the effect of *Mettl3* knockdown on global protein translation in the adult fly brain using the puromycin assay to label nascent protein synthesis in vivo with 24 h of puromycin incorporation[51]. Analysis of *Mettl3* knockdown brains showed increased puromycin incorporation compared to control brains, indicating that *Mettl3* RNAi globally promotes protein synthesis in the brain (Fig. 6a, left). A reciprocal significant decrease in translation occurred in the brain upon *Mettl3* upregulation (Fig. 6a, right). Upon HS, *Mettl3* knockdown brains maintained higher puromycin incorporation than brains with *Mettl3* knockdown or HS exposure alone (Fig. 6b). *Mettl3* ΔCat mutant/*Mettl3* deficiency brains, and *Mettl14* RNAi brains similarly showed an increase in puromycin levels compared to controls (Supplementary Fig. 9a–c). Treatment of puromycin plus actinomycin D (a transcription inhibitor) consistently showed increased puromycin incorporation in *Mettl3* knockdown brains (Supplementary Fig. 9d). Together, these results indicate that *Mettl3* gene function modulates global protein translation in the brain in vivo.

To probe the possible downstream processing of m⁶A transcripts, we focused on understanding the impact of the two YTH m⁶A reader proteins: cytoplasmic Ythdf and nuclear Ythdc1. Knockdown of either *Ythdf* or *Ythdc1* showed increased puromycin incorporation in the brain (Fig. 6c), and upregulation yielded the reverse (Fig. 6d). These results support *Ythdf* and *Ythdc1* as key regulators of translation of m⁶A targets in the brain.

## *Ythdc1* knockdown animals are resilient to heat stress

To determine the YTH m⁶A reader protein involved in the brain's stress response downstream of *Mettl3*, we examined whether HS or *Mettl3* loss influenced the transcript levels of *Ythdf* or *Ythdc1*. Brain RNA-Seq data indicated that *Ythdc1* transcript levels were increased in control brains upon HS, in an effect that lasted 6 h post-HS (Fig. 7a and Supplementary Data 5). Surprisingly, *Mettl3* knockdown attenuated this stress-induced feature of *Ythdc1*, and *Mettl3* RNAi brains had reduced *Ythdc1* levels compared to control in basal and heat shock conditions (Fig. 7a, b). The protein levels of *Ythdc1* were similarly impacted by *Mettl3* knockdown and by HS (Fig. 7b and Supplementary Fig. 9e). By contrast, *Ythdf* transcript levels were minimally impacted in control, HS, or with *Mettl3* knockdown (Supplementary Data 5 and Supplementary Data 8). Due to the changes in *Ythdc1* transcript and protein levels induced with HS and with *Mettl3* knockdown in the brain, we focused on the biological effects of *Ythdc1* perturbation.

We first assessed whether the knockdown of *Ythdc1* influenced the animals' heat stress resilience. *Ythdc1* knockdown conferred heat

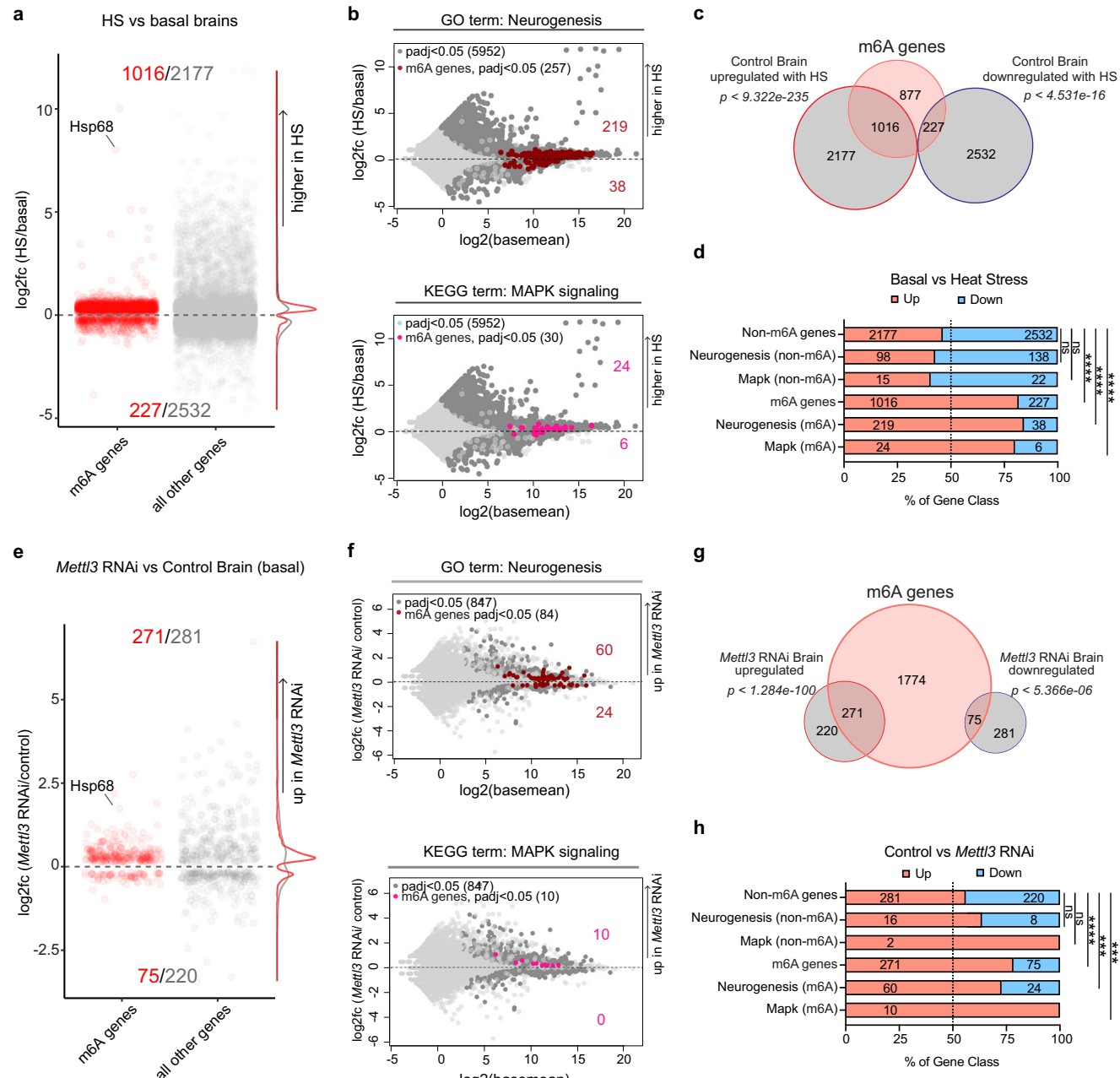

**Fig. 4 | m⁶A transcripts are increased with HS and upon *Mettl3* knockdown.**
**a** Plot of significantly differentially expressed genes in HS vs basal of control brains (DaGal4 > mCherry RNAi). Positive logFC indicates an increase in transcript levels with HS. m⁶A genes (red), or all other genes (gray). All RNA-seq differential expression analysis are provided in Supplementary Data 5. **b** MA plots of *Mettl3*-dependent m⁶A genes that are enriched in the neurogenesis (GO term) and MAPK (KEGG pathway). Plots show control brains HS vs basal (positive logFC indicates an increase with HS brains). **c** Venn diagram of m⁶A genes that are up or downregulated upon heat stress. Hypergeometric test, one-sided. $p < 9.322\text{e-}235$, $p < 4.531\text{e-}16$. **d** Percentage of indicated gene classes significantly upregulated (red) or downregulated (blue) in basal versus HS brains RNA-seq. ****$p < 0.0001$ from one-sided Fisher's test. $p = 0.0003$, $p < 2.2\text{e-}16$, $p < 2.2\text{e-}16$, ns not significant. **e** Plot of significantly differentially expressed genes in DaGal4 > mCherry RNAi vs *Mettl3* RNAi brains in basal condition. Positive logFC indicates an increase in transcript levels in *Mettl3* RNAi brains. m⁶A genes (red), or all other genes (gray). All RNA-seq differential expression analysis are provided in Supplementary Data 5. **f** MA plots of m⁶A genes that are enriched in the neurogenesis (GO term) and MAPK (KEGG) pathways. Plots show logFC of *Mettl3* RNAi versus control brains (positive logFC indicates an increase in *Mettl3* RNAi brains). **g** Venn diagram of m⁶A genes that are up or downregulated upon *Mettl3* knockdown. Hypergeometric test, one-sided. $p < 1.284\text{e-}100$, $p < 5.366\text{e-}06$. **h** Percentage of indicated gene classes significantly upregulated (red) or downregulated (blue) in control versus *Mettl3* RNAi brains RNA-seq. ***$p < 0.001$, ****$p < 0.0001$ from one-sided Fisher's test. $p = 0.006225$, $p = 0.008509$, $p = 1.799\text{e-}11$. Source data and statistical analysis are provided as a Source Data file.

stress resilience (Fig. 7c), similar to the effect of *Mettl3* knockdown, while upregulation resulted in increased sensitivity to heat stress (Fig. 7d). We then performed RNA-Seq from control and *Ythdc1* RNAi brains in basal and HS conditions. We examined the genes that were differentially expressed upon *Ythdc1* RNAi (Supplementary Data 5). Sixty-five genes upregulated in *Ythdc1* knockdown brains at baseline overlapped genes upregulated with *Mettl3* knockdown (Fig. 7e). Many of these genes were key HS chaperones and marked by m⁶A and upregulated by *Mettl3* knockdown (*Hsp70, DnaJ-1, Hsp68*) (Fig. 7e). GO term analysis of all upregulated transcripts in *Ythdc1* RNAi brains showed enrichment in heat shock protein chaperone pathways (Fig. 7f). GO term analysis of genes marked by m⁶A and upregulated

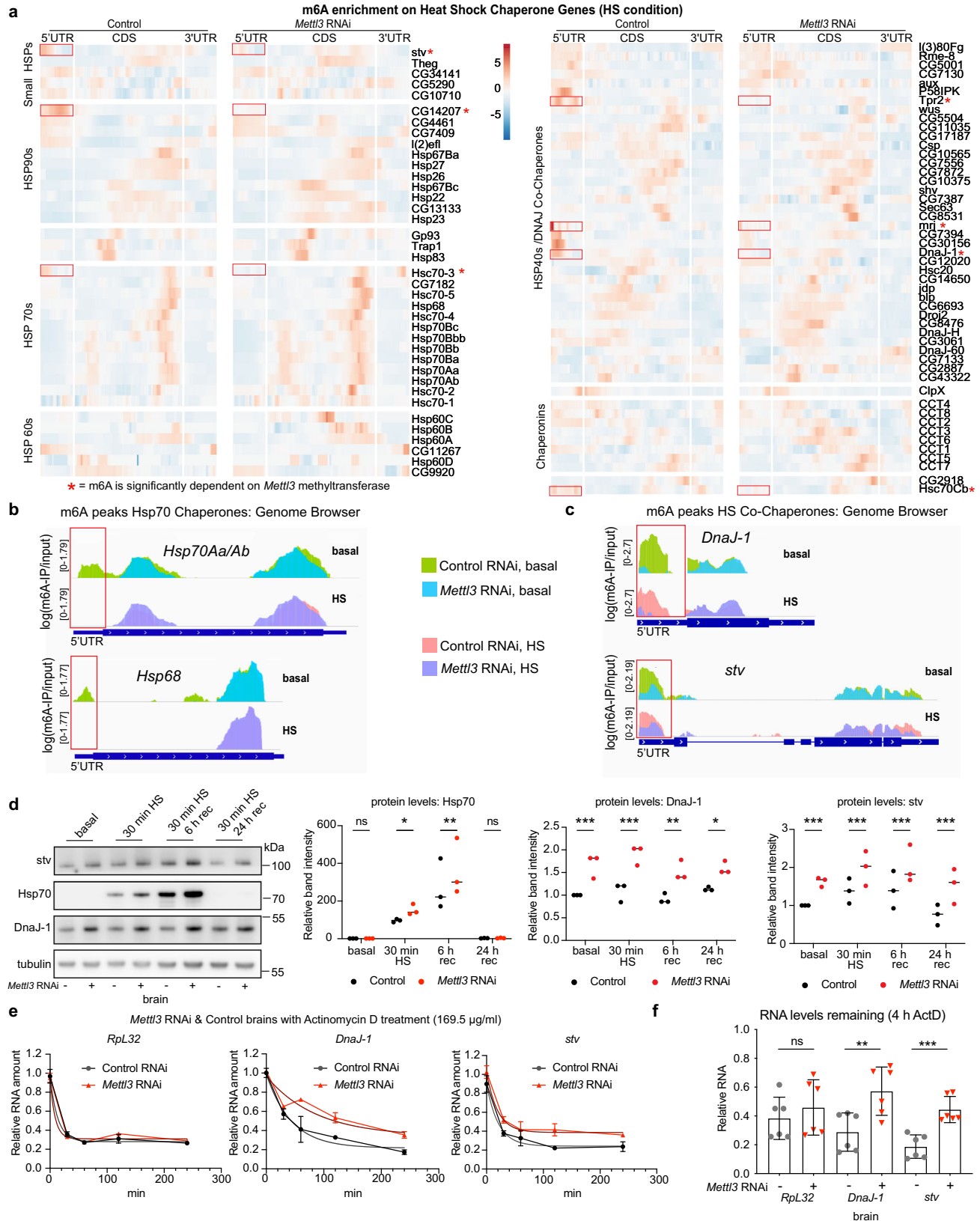

★ = m6A is significantly dependent on *Mettl3* methyltransferase

with *Ythdc1* RNAi showed enrichment in signaling and neurogenesis terms (Fig. 7f), pathways we observed being increased with HS (Fig. 4). We confirmed that *Ythdc1* RNAi, like *Mettl3* RNAi, showed increased levels of Hsp70 protein at baseline and after HS compared to control brains (Fig. 7g). Thus, *Ythdc1* knockdown in the brain resulted in both increased levels of heat stress chaperone genes and animal stress

resilience, both of which are similar to *Mettl3* knockdown, suggesting that *Ythdc1* plays a role in the m⁶A HS response of the brain.

## Discussion

m⁶A regulates diverse critical physiological processes, including neurodevelopment, immune responses, cancer metastasis, learning,

**Fig. 5 | *Mettl3* knockdown increases select HS chaperone protein levels and decreases their rate of RNA decay. a** Heat map of m⁶A enrichment of HS Chaperone Genes (Supplementary Data 6). Shown is control and *Mettl3* RNAi m⁶A enrichment in HS conditions. m⁶A enrichment is shown as m⁶A-IP divided by the input. Heat map displays *z*-score values, which are scaled by row, each gene is relative to itself and relative across all six boxes. Segmented into 5′UTR, CDS, and 3′ UTR. **b** Example tracks from genome browser of m⁶A locations for HS chaperone transcripts *Hsp70Aa/Ab* and *Hsp68* in basal and HS 30 min, from control and *Mettl3* RNAi brains. Tracks shown as log(m⁶A-IP divided by input). Separate IP and Input tracks are shown in Supplementary Data 7c. **c** Example tracks from genome browser of m⁶A locations for m⁶A marked HS co-chaperone genes *DnaJ-1* and *stv* in basal and HS from control and *Mettl3* RNAi brains. **d** Hsp70, DnaJ-1, and stv protein levels from DaGal4 > *Mettl3* RNAi vs DaGal4 > mCherry RNAi brains dissected in basal, HS 30 min at 38.5 °C, 6 h recovery post-HS, or 24 h recovery post-HS. Right is the quantification of three biological replicate immunoblots, 15 brains per replicate.

Data presented as mean, *$p < 0.05$, **$p < 0.01$, ***$p < 0.001$, two-way ANOVA with Sidak's test. In **d** Hsp70, ns not significant, $p = 0.0383$, $p = 0.0027$, DnaJ-1, $p = 0.0009$, $p = 0.0001$, $p = 0.0022$, $p = 0.0196$, stv, $p = 0.0005$, $p = 0.0006$, $p = 0.0005$, $p = 0.0001$. **e** Actinomycin D RNA decay assay from mCherry RNAi and *Mettl3* RNAi fly brains dissected and incubated in Schneider's *Drosophila* Medium plus 169.5 ug/ml of actinomycin D for 30 min, 1, 2, 4 h, and control with no actinomycin D. RNA was extracted and used for RT-qPCR to determine relative RNA levels of *RpL32*, *DnaJ-1*, and *stv*. Data presented as mean ± SD of one biological replicate with three technical replicates. Gray line is best-fit curve for mCherry RNAi, and the red line is best-fit curve for *Mettl3* RNAi. **f** Relative RNA levels remaining at 4 h of actinomycin D treatment. $n = 6$ biological replicates, 15 brains per replicate. Data presented as mean ± SD, **$p < 0.01$, ***$p < 0.001$, Student's two-tailed *t*-test. *Rpl32*, ns not significant, *DnaJ-1*, $p = 0.0088$, *stv*, $p = 0.0004$. Source data and statistical analysis are provided as a Source Data file.

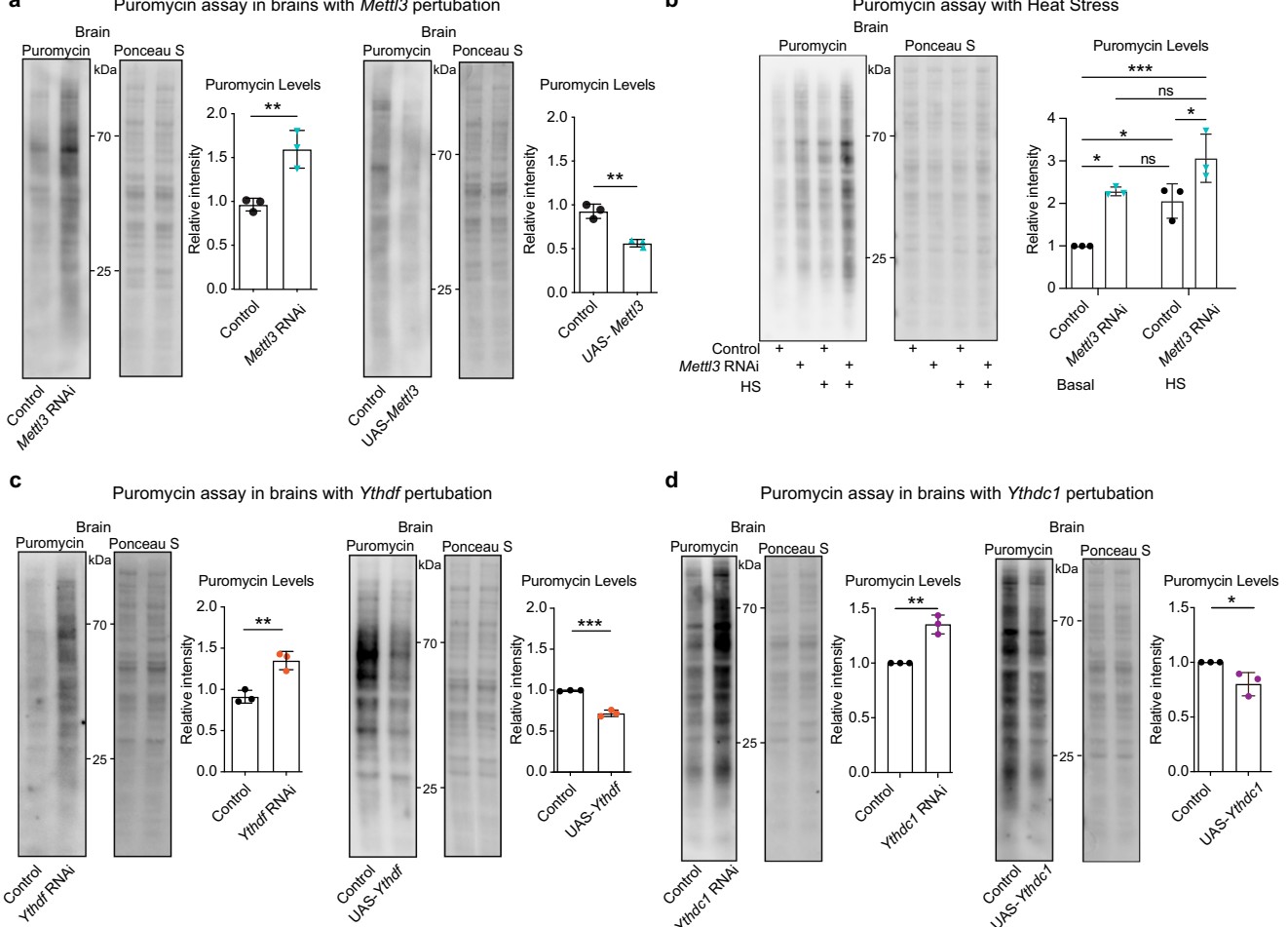

**Fig. 6 | *Mettl3* attenuates protein translation in the brain. a** Control, *Mettl3* RNAi, and *UAS-Mettl3* upregulation flies were fed 600 uM puromycin for 24 h. (Genotypes: DaGal4 > mCherry RNAi vs DaGal4 > *Mettl3* RNAi, ElavGS > UAS-mCherry vs ElavGS > *UAS-Mettl3*). Brains were dissected from the heads, and protein immunoblots were probed for puromycin incorporation in each sample. Ponceau S staining shows total protein levels. $n = 3$ biological replicates, 15 brains per replicate. Data presented as mean ± SD, *$p < 0.05$, **$p < 0.01$, Student's two-tailed *t*-test. $p = 0.0085$, $p = 0.0024$. **b** Control and *Mettl3* RNAi flies were HS for 30 min and then fed 600 uM puromycin for 4 h or control and fed puromycin for 4 h. (Genotypes: DaGal4 > mCherry RNAi vs DaGal4 > *Mettl3* RNAi). Brains were dissected after 4 h, and protein immunoblots were probed for puromycin levels in

each sample. Ponceau S staining shows total protein levels. $n = 3$ biological replicates, 15 brains per replicate. Data presented as mean ± SD, *$p < 0.05$, ***$p < 0.001$, two-way ANOVA with Sidak's test. $p = 0.0125$, $p = 0.0367$, $p = 0.0006$, $p = 0.0476$, ns not significant. **c** ElavGS > mCherry RNAi vs *Ythdf* RNAi, and ElavGS > UAS-mCherry vs *UAS-Ythdf* upregulation flies brain puromycin assay. $n = 3$ biological replicates, 15 brains per replicate. Data presented as mean ± SD, **$p < 0.01$, Student's two-tailed *t*-test. $p = 0.0049$, $p = 0.0003$. **d** ElavGS > mCherry RNAi vs *Ythdc1* RNAi, and ElavGS > UAS-mCherry vs *UAS-Ythdc1* upregulation flies brain puromycin assay. $n = 3$ biological replicates, 15 brains per replicate. Data presented as mean ± SD, **$p < 0.01$, Student's two-tailed *t*-test. $p = 0.0021$, $p = 0.0311$. Source data and statistical analysis are provided as a Source Data file.

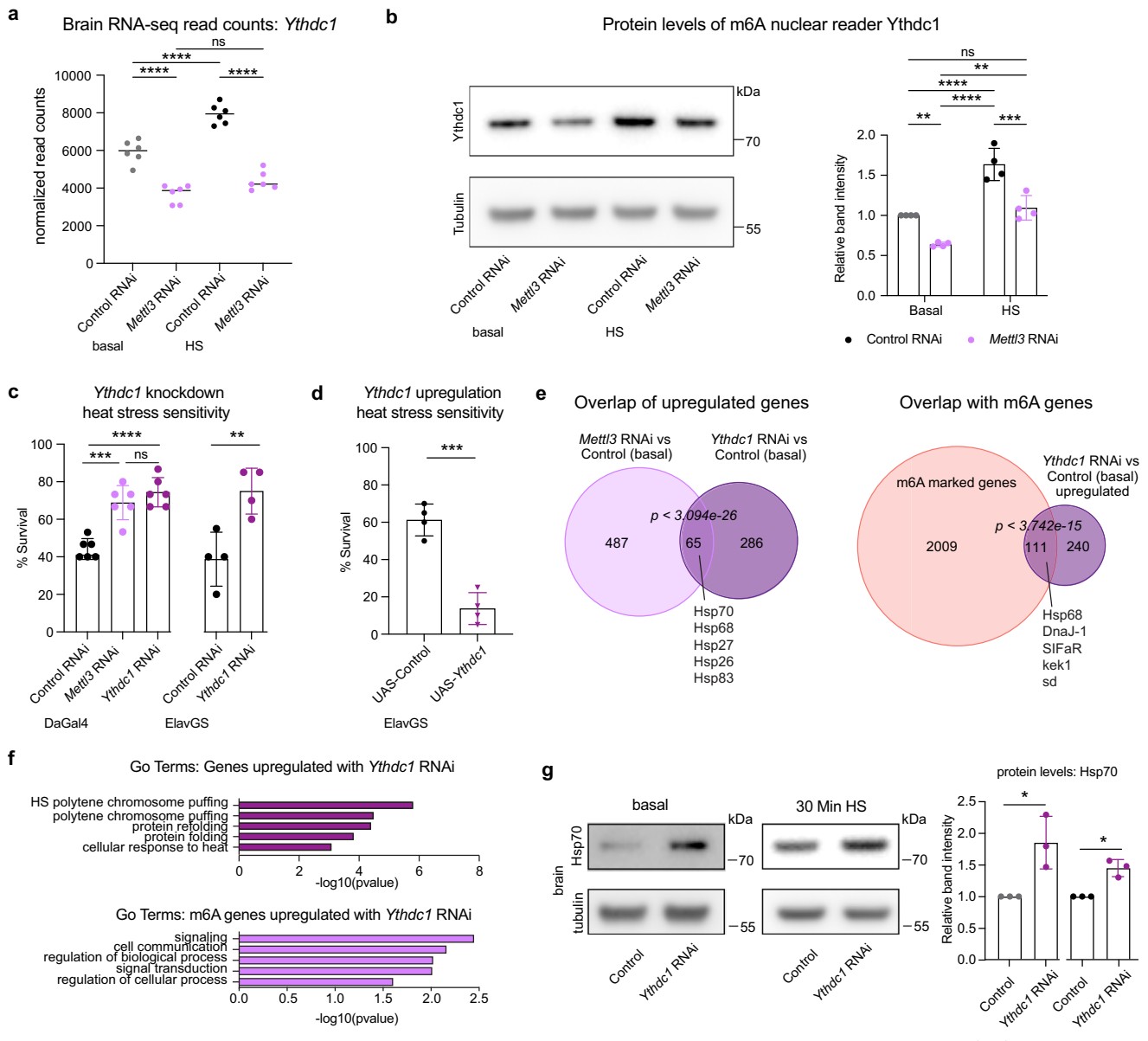

**Fig. 7 | m⁶A reader protein *Ythdc1* knockdown animals are resilient to heat stress. a** Normalized read counts of *Ythdc1* from brain RNA-sequencing in baseline, HS, Control, and *Mettl3* RNAi conditions (Dagal4 > mCherry RNAi vs *Mettl3* RNAi). Statistical analysis from RNA-seq differential expression analysis, provided in Supplementary Data 5. ****$p < 0.0001$ From left to right $p = 1.53E\text{-}11$, $p = 5.59E\text{-}07$, $p = 1.71E\text{-}38$, ns not significant. **b** Ythdc1 protein levels from Dagal4 > mCherry RNAi vs *Mettl3* RNAi brains dissected in basal or HS 30 min at 38.5 °C. $n = 4$, 15 brains per replicate. Quantification of four biological immunoblots show decreased expression of Ythdc1 protein in *Mettl3* RNAi fly brain and increase of Ythdc1 protein with HS. Data presented as mean ± SD, **$p < 0.01$, ***$p < 0.001$, ****$p < 0.0001$. Student's two-tailed *t*-test. $p = 0.0078$, $p < 0.0001$, $p < 0.0001$, $p = 0.0013$, $p = 0.0003$. **c** Ubiquitous knockdown of *Ythdc1 or Mettl3* (DaGal4), and neuron-specific knockdown (ElavGS), and corresponding controls (mCherry RNAi) were HS for 1.5 h at 38.5 °C and scored for survival after 24 h recovery. $n = 6$ biological replicates for DaGal4, $n = 4$ biological replicates for ElavGS. Each data point represents percent survival in a vial of 20 flies per replicate. Data presented as mean ± SD, **$p < 0.01$, ***$p < 0.001$, ****$p < 0.0001$, One-way ANOVA, Student's two-tailed *t*-test. $p = 0.0002$, $p < 0.0001$, ns not significant, $p = 0.0085$. **d** Neuron-

specific (ElavGS) upregulation of *Ythdc1* and control were HSed for 1 h at 38.5 °C and scored for survival after 24 h recovery. $n = 4$ biological replicates. Each data point represents percent survival in vial of 20 flies per replicate. Data presented as mean ± SD, ****$p < 0.0001$, Student's two-tailed *t*-test, $p = 0.0002$. **e** Overlap of genes upregulated upon Dagal4 > *Mettl3* RNAi and *Ythdc1* RNAi (baseline) and overlap of m⁶A genes and genes upregulated with *Ythdc1* RNAi. Differential expression analysis is provided in Supplementary Data 5. Hypergeometric test, one-sided, $p < 3.093e\text{-}26$, $p < 3.742e\text{-}15$. **f** GO term enrichment of genes upregulated upon *Ythdc1* RNAi (baseline). Go terms of m⁶A genes upregulated upon *Ythdc1* RNAi. The -log10(*p*-val) enrichment of genes in each category is shown. Supplementary Data 3 for full GO term list and *p* values. **g** Hsp70 protein levels from DaGal4 > mCherry RNAi and *Ythdc1* RNAi brains dissected in basal or HS 30 min at 38.5 °C. $n = 3$, 15 brains per replicate. Quantification of three biological immunoblots, 15 brains per replicate, show increased expression of Hsp70 protein in *Ythdc1* RNAi fly brain. Data presented as mean ± SD, *$p < 0.05$, Student's two-tailed *t*-test. $p = 0.0237$, $p = 0.0126$. Source data and statistical analysis are provided as a Source Data file.

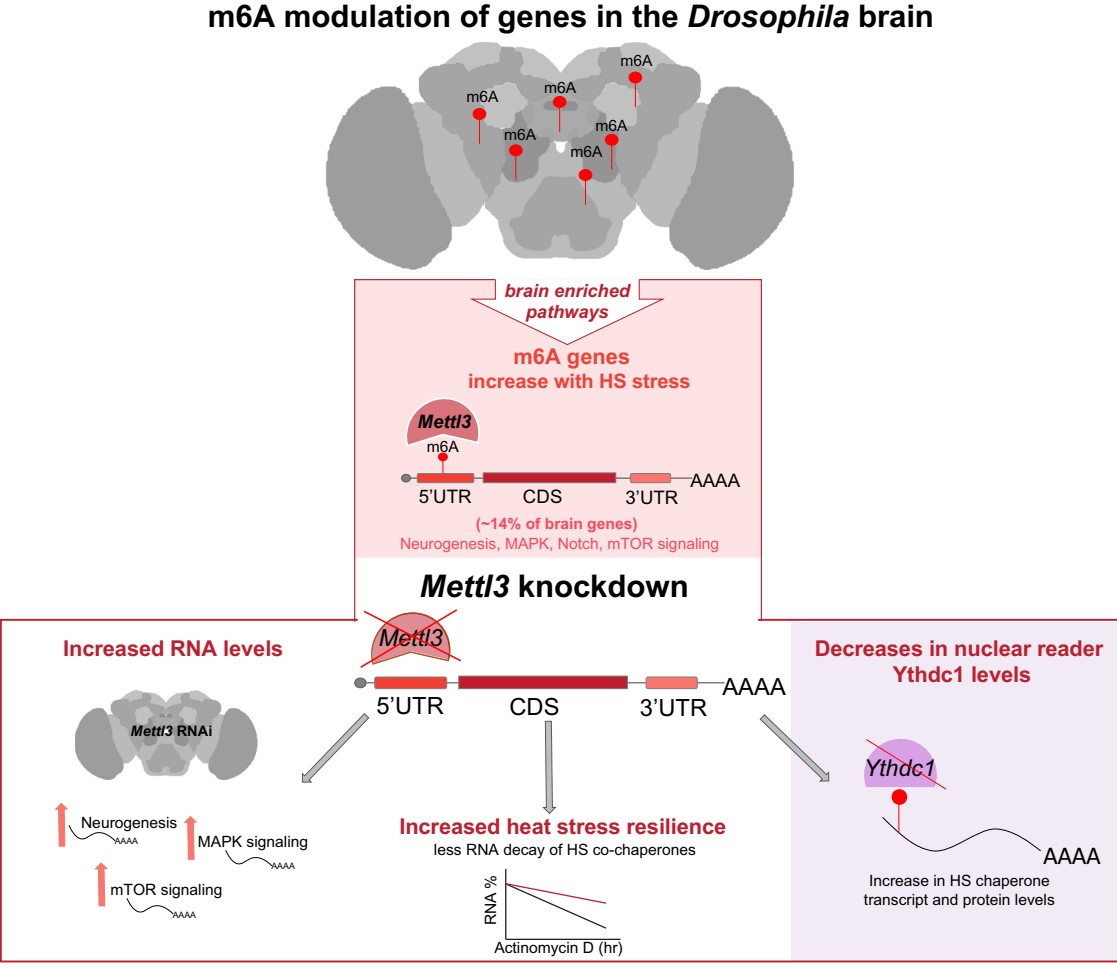

**Fig. 8 | m⁶A modulation of genes in the *Drosophila* brain.** Our data indicate a model whereby m⁶A is enriched in the brain and in the 5′UTR of select transcripts in the *Drosophila* brain. m⁶A modification on polyA+ RNA shows a dynamic increase in the brain with acute heat stress. Levels of m⁶A modified gene transcripts that increase with heat stress are enriched for neurogenesis and dynamic signaling pathways. *Mettl3* knockdown decreases the level of m⁶A in the 5′UTR of its target transcripts and biologically results in increased heat stress resilience of the animal, and a brain-specific increase in protein and RNA levels of m⁶A targets, with decreased RNA decay. *Mettl3* knockdown also leads to a decrease in transcript and protein levels of Ythdc1. *Ythdc1* knockdown is associated with an increase in key HS chaperone transcripts and protein levels, and stress resilience.

memory, and stress[18,19]. Here, we provide insight into the role of m⁶A in the adult *Drosophila* brain in vivo in an acute stress situation: heat shock. m⁶A modification of polyA+ RNA shows a brain-enriched *Mettl3*-dependent increase with stress. Decreasing m⁶A levels with knockdown of *Mettl3* confers stress resistance. m⁶A is enriched in the fly brain and in the 5′UTR of polyA+ transcripts with neuronal and signaling function. Knockdown of reader protein *Ythdc1* showed similar stress resilience and an increase in key HS chaperone levels. Together, these data suggest a critical involvement of m⁶A in the modulation of protein and RNA levels in the brain in response to stress (Fig. 8).

### The impact of m⁶A on the brain stress response

m⁶A likely contributes to the surprisingly distinct response of the brain to acute HS in *Drosophila*. While the brain and head both have a robust increase in *Hsp70* transcription following HS, in the brain, Hsp70 protein levels are only mildly upregulated compared to heads (see Supplementary Fig. 2c). *DnaJ-1* and *stv*, which promote *Hsp70* function[44,48], also had a robust transcriptional response to HS in brains, heads, and whole fly tissue, but strikingly their protein levels

were not well upregulated with HS in brain tissue. *Mettl3* knockdown increased stress-induced protein levels of Hsp70, DnaJ-1, and stv, suggesting that 5′UTR m⁶A dampens the levels of key stress chaperones associated with the acute HS response in the *Drosophila* brain. Furthermore, *Mettl3* and *Ythdc1* knockdown elevated the levels of these key HS chaperones, mimicking mild preconditioning stress; preconditioning is known to be crucial for an organism's response to severe stress[52]. Conversely, upregulation of *Mettl3* or *Ythdc1* increased stress sensitivity.

In mammalian cells, m⁶A has been shown to be critical to the cap-independent translation of *Hsp70* with HS[26,27,53]. However, in mammalian cells *Mettl3* (along with microprocessor *Dgcr8*) has also been shown to co-transcriptionally mark *Hsp70*, leading to RNA degradation[50]. In our in vivo *Drosophila* brain data, stress-induced *Hsp70* transcripts were not m⁶A modified in the 5′UTR upon HS. By contrast, we observed basal m⁶A enrichment in the 5′UTR of *Hsp70* and *Hsp68* in control flies that was lost upon HS and with *Mettl3* knockdown (see Fig. 5b, c and Supplementary Fig. 7b). Brain RNA-seq analysis showed that both *Hsp70* and *Hsp68* RNA levels were elevated at

baseline in the *Mettl3* RNAi brain, and *Mettl3* knockdown increased Hsp70 protein levels in the brain. Taken together, these data suggest that m[6]A normally dampens the transcript and protein levels of the Hsp70 class of chaperones (stress-induced Hsp70s and Hsp68) in the *Drosophila* brain. Consistent with a previous study in mammalian cells[50], our data (Fig. 5e, f) indicate that m[6]A is involved in RNA decay of stress-inducible chaperones in the brain in vivo.

Phenotypically, *Mettl3* knockdown resulted in increased stress resilience. m[6]A genes are enriched for many critical signaling pathways, such as MAPK, Hippo, WNT, mTOR and Notch signaling, and our data indicate transcripts for these pathways increase with HS in the brain (Fig. 3e). These signaling pathways have not been previously implicated in the canonical HS response, although they are employed by neurons to deal with various types of stressful situations, such as proteostasis and aging[10,11]. By dot blot, mass spec and m[6]A-IP, our data indicate an increased level of m[6]A with HS. We underscore that our RNA-seq data are from dissected adult fly brain tissue vs heads or cultured cells. In the brain, m[6]A targets are upregulated upon both HS and *Mettl3* RNAi, and the increase with HS may account for the overall increase in m[6]A levels we observe upon HS. The increased transcript levels of m[6]A genes, and higher levels of critical chaperones Hsp70 and DnaJ-1, may contribute to the increased stress resilience of the animals upon *Mettl3* knockdown. We suggest that m[6]A modulation of genes normally leads to an attenuated acute stress response in the brain, given the dampened levels of these m[6]A-modified chaperone genes in the brain compared to other tissues. This may underlie, at least in part, the unique susceptibility of the brain to severe stresses like protein misfolding diseases and dementia with aging.

Due to their postmitotic state, neurons often utilize altered molecular mechanisms, and perhaps m[6]A is utilized in neuronal cells to tightly control neuronal signaling and chronic stress pathways, with the caveat of a dampened acute stress response. In our data, we find that m[6]A targets are more highly expressed in brain tissue compared to epithelial S2 cells (Supplementary Fig. 5h and Supplementary Data 4). This suggests that regulation of m[6]A is best examined in the tissue type of interest where the m[6]A targets are expressed. Landscape m[6]A methylome analysis in human and mouse tissues has recently shown that brain tissue methylomes are highly specific and correlate with the relative expression levels of m[6]A complex reader and writers[54].

## The molecular role of m[6]A on transcript and protein levels

Some studies have shown m[6]A associated with stress translation initiation[26,35], although additional studies point towards m[6]A association with translation inhibition[24,36]. m[6]A-modified transcripts are regulated through m[6]A binding proteins, and can be directed to specific compartments of the cell for stability and translation. RNA reader proteins may sequester RNAs during stress, such as YTHDF shunting of m[6]A-marked transcripts into P-bodies and stress granules[32,33]. This may prevent the translation of m[6]A transcripts and promote their degradation. Furthermore, recent studies have uncovered additional m[6]A binding factors, such as Fmr1, which suppresses the translation of its targets[34,40]. In *Drosophila*, most m[6]A sites reside in the 5′UTR (this study and [39]), which might suggest a role in translation regulation[26,35]. Additional studies suggest that m[6]A is preferentially deposited on fly transcripts with lower translational efficiency[39]. In the *Drosophila* brain, our data suggest that decreased *Mettl3* promotes increased protein translation and increased RNA levels of targets. Knockdown of m[6]A reader proteins *Ythdc1* and *Ythdf* both led to increased nascent protein synthesis in the brain. Thus, these two reader proteins appear to lead to repressed translation. Although *Mettl3* has functions other than serving in the methyltransferase complex, including chromatin regulation[55] and modifies RNAs beyond mRNAs (enhancer RNAs, circRNAs, and lncRNAs)[56,57], we found that HS resilience and protein translation in the brain are dependent on both *Mettl3* and *Mettl14* (see Fig. 2d, Supplementary Fig. 3, Supplementary Fig. 9).

Previous studies in mammalian cells have shown that YTHDF2 is the m[6]A reader protein that increases upon heat stress[27]. Our in vivo brain RNA-sequencing revealed that nuclear m[6]A reader *Ythdc1* levels are increased upon heat stress in the fly brain and that knockdown of *Mettl3* strikingly decreases levels of *Ythdc1*. Functional analyses revealed that knockdown of *Ythdc1* was beneficial to the brain stress response in vivo, with the animals being more resilient to heat stress. The increase in *Ythdc1* levels upon HS in wild-type brains may serve as insight into mechanisms the brain has in place that abrogate the acute stress response. Our data indicate that knockdown of *Ythdc1* up-regulates m[6]A target chaperones such as *Hsp68* and *DnaJ-1*, as well as signaling pathway genes, which may be critical to the HS recovery response. Given the enrichment of m[6]A-modified genes in signaling pathways, the role of m[6]A in the brain may be to allow greater control of gene players in chronic stress signaling, but with a compromise in response to acute stress.

Our work expands the understanding of the role of m[6]A in the brain and in the acute stress response. We suggest that m[6]A genes—enriched in critical neuronal and signaling pathways—may require exquisite dynamic control: m[6]A modification may be one mechanism that the brain uses uniquely to exert that control. m[6]A deposition thus modulates neuronal signaling and brain-enriched stress response pathways by fine-tuning RNA translation and decay in the brain. We speculate that m[6]A modification of these genes benefits the overall long-term maintenance of brain function by adding a greater level of dynamic regulation of those genes. However, this appears to come at the cost of decreased resilience of the brain to acute stress, and perhaps also contributes to the selective vulnerability of the brain to neurodegenerative disease.

## Methods

### *Drosophila* work and lines

A full list of *Drosophila* stocks used in this study are described in Supplementary Data 7. RNAi lines were generated by the Harvard Transgenic RNAi Project (TRiP)[58], and stocks were obtained from the Bloomington *Drosophila* stock center, Indiana, USA. Crosses were performed at 25° and grown on standard cornmeal molasses agar. Driver lines used: DaGAL4, DaGS(GeneSwitch)-GAL4, elav(Geneswitch)-GAL4 as indicated per experiment. For conditional expression using the Gal4-GS(GeneSwitch) system, flies were collected 1–2 days after eclosion and placed on food vials pre-coated with 100 ul of 4 mg/mL RU486 per vial for 6 days before stress experiments. Food is changed every second day. (Mifepristone, Sigma-Aldrich). For all experiments, male flies were used for consistency in the experiments and to avoid issues in RU486 food due to egg laying of females.

### Brain dissections

Brain dissections were conducted as previously described in[59]. Briefly, flies were anesthetized using $CO_2$ and decapitated using forceps. The head was placed posterior side down and the proboscis was then removed using Dumont #5SF forceps (Fine Science Tools, 11254-20). The brain was then gently popped out through the proboscis cavity, cleaned in PBS, and transferred to an RNAse-free microfuge tube and PBS was aspirated. Brains were then ground in Laemmli Buffer (5 μL per brain, at least 10–20 brains for each sample) for Western immunoblotting or Trizol for RNA analysis.

### Western immunoblot analysis

Brain or head samples were homogenized in sample buffer of 1x Laemmli sample Buffer (Bio-rad, 1610737), 50 ul b-mercaptoethanol (Sigma, m6250), 1x protease inhibitor (Roche, 11836170001), and 1 mM PMSF (Sigma, P7626). About 5 ul of sample buffer is added per brain, 7.5 ul added per head, and 40 ul added per whole fly. Samples are boiled at 98 °C for 3 min and then centrifuged at 1500 rpm for 3 min at room temperature. Sample was loaded onto 15 wells of 1.0 mm 4–12%

Bis-Tris NuPAGE gels (Thermo Fisher, WG1401) with a pre-stained protein ladder (Thermo Scientific, 22619). 1 brain, 1 head, or 8% of whole fly tissue is loaded on each lane per experiment. Gel electrophoresis was performed using Xcell Surelock Mini-Cell Electrophoresis System at 140 V and transferred overnight onto a nitrocellulose membrane 0.45 µM (Bio-rad, 1620115), using a Bio-rad mini transblot cell at 90 A for 16 h. Membranes were stained in Ponceau S (Sigma, P7170-1L), washed in DI water, and imaged with Amersham Imager 600. Ponceau S was washed off in 3 × 5 min in Tris-buffered saline with 0.1% Tween20 (TBST). The membrane was blocked in 5% non-fat dry milk (LabScientific, M08410) in TBST for 1 h, and incubated with primary antibodies with blocking buffer overnight at 4 °C. Following 3 × 5 min washed in TBST, membranes were incubated with HRP-conjugated secondary antibodies at 1:5000 for 1 h at room temperature in blocking solution. Membranes were washed 3 × 5 min in TBST and the signal was developed using ECL prime (Cytivia, RPN2232) and detected using an Amersham Imager 600. Primary antibodies used: anti-tubulin (1:5000, DHSB, AA4.3, 5/31/18−44 ug/ml), anti-Hsp70 (1:5000, Sigma, 7FB-SAB5200204-100uG,141002), anti-Mettl3 (1:5000, Proteintech, 15073-1-AP, Ag7110), anti-HSP40 (1:5000, Enzo Life Sciences, ADI-SPA-400-D,04062141), anti-stv (1:5000, Proteintech, 13913-1-AP, Ag4905, validation included in Source Data File), anti-fl(2)d (1:10, DSHB-9G2, 10/18/18−42 ug/ml), anti-futsch (1:600, DSHB-22C10, 10/10/19−53 ug/ml), anti-drpr (1:400, DSHB-5D14, 6/22/17−36 ug/ul), anti-puromycin (1:1,000, Kerafast, EQ0001, 200517), anti-hsf (1:20:000, anti-rabbit, gift from John Lis[60,61]). Rabbit anti-Ythdc1 (1:5000), affinity purified rabbit antibody created by Vivitide against 18 residues of Ythdc1 (157-173 "CRTKIPSNANDSAGHKSD"). Secondary Antibodies used: Goat anti-mouse (1:5000, Jackson lmmunoResearch, 115-035-146,153978), Goat anti-Rabbit (1:5000, Jackson lmmunoResearch, 111-035-144, 138306), Goat anti-rat (1:5000, Thermo Fisher Scientific, A10549, 2273679).

## RNA extraction

Tissue was homogenized in 200 ul of Trizol (Thermo Fischer Scientific, 15596026) in RNase-free 1.5 ml microfuge tubes (Thermo Fischer Scientific, AM12400). About 800 ul of Trizol (Thermo Fischer Scientific, 15596026) was added to the tube and 200 ul of chloroform (Fisher Scientific, AC423555000) and was vigorously shaken for 20 s at room temperature. Samples were left for 5 min at RT to form the upper aqueous phase and centrifuged at 4 °C for 15 min at 12,000 × g. The upper aqueous phase was transferred to a fresh RNase-free tube. For head tissue samples, RNA was precipitated in 1 vol of isopropanol (Fisher Scientific, ICN19400690) and 1/10th vol 3 M sodium acetate (Thermo Fischer Scientific, AM9740) and left at −80 overnight. Samples were centrifuged for 30 min at 21,000 × g at 4 °C, the RNA pellet was washed in 70% ethanol, centrifuged for 10 min at 21,000 × g at 4 °C, air-dried, and resuspended in 50 ul of RNAse-free DEPC treated water (Ambion, AM9906). Genomic DNA in was digested with turbo DNase (Thermo Fischer Scientific, AM2238) using the vigorous protocol. Brain RNA samples were processed using the Zymo RNA clean & concentrator −5 kit (Zymo, R1013), using their RNA clean-up from the aqueous phase after Trizol /chloroform extraction protocol plus on-column DNaseI treatment. RNA amount was measured using a nanodrop, and integrity was validated by an Agilent 2100 Bioanalyzer using an RNA nano chip.

## Real-time PCR

About 400 ng RNA was used per cDNA reaction using the High Capacity cDNA Reverse Transcription Kit (Applied Biosystems, Thermo Fisher Scientific, 4368814). cDNA was then used for qPCR reactions set up with SYBR Green Fast Reagents, using 384-well plates on the Applied Biosystems ViiA7 machine. Primers used are in Supplementary Data 7. Mean fold-change was determined using the ΔΔCt method. Each experiment used technical triplicates as well as three

biological replicates; Graphpad prism 8/9 software was used for statistics.

## Stress sensitivity assay

Fly crosses were carried out at 25 °C. Adult flies were collected and aged to 6 days post eclosion or after 6 days on RU food. Flies were anesthetized and transferred to clear plastic 13 ml vials, and cotton was placed at the 4 ml mark on the vials to concentrate the flies near the bottom. Each vial contained 20 flies. Flies were allowed to recover for 30 min and then transferred to a water bath for mild non-lethal heat stress (30 min at 38.5 °C) or a longer heat shock (1−1.5 h at 38.5 °C) for a severe stressor to measure stress sensitivity survival. The flies were then transferred to normal food and allowed to recover overnight at 25 °C. After recovery, the percent of flies alive versus dead was recorded per vial.

## LC-MS/MS analysis of m⁶A Levels

PolyA+ RNA was extracted from brains and heads using the NEBNext Poly(A) mRNA Magnetic Isolation Module (NEB, E7490L). LC-MS/MS was conducted as previously described in ref. 62. All quantifications were performed by using the standard curve obtained from pure nucleoside standards running with the same group of samples. Then, the percentage ratio of m⁶A to A was used to compare the different modification levels.

## Dot blot assay

Total RNA was collected from heads or whole flies using standard trizol chloroform extraction. $n = 200$ head per condition or $n = 10$ whole flies per condition. PolyA+ RNA was obtained using Dynabeads mRNA Direct Purification Kit (Ambion, 61011). Dots (total RNA or polyA+ RNA) were applied to an Amersham Hybond-N⁺ membrane (GE Healthcare, RPN119B) in duplicate as 100 ng RNA per 1 ul dot. Dots were done on a Dry Membrane in a clean petri dish. The membrane was completely dried before RNA was crosslinked to the membrane using a UV Stratalinker 2400 by running the auto-crosslink program twice. The membrane was then washed in PBST three times 5 min each, blocked with 5% non-fat milk in PBST for 2 h. The membrane was incubated with primary anti-m⁶A antibody (1:1,000, Synaptic Systems, 202003, 2−97) overnight at 4°, then washed in PBST 3 × 5 min, incubated in HRP-conjugated anti-rabbit IgG secondary antibody (1:5000) for 2 h at room temperature, washed in PBST 3 × 5 min, and visualized using ECL prime. The membrane is washed in PBST and then incubated in methylene blue for 15 min, rinsed in PBST and imaged.

## M⁶A-IP sequencing

Total RNA was extracted from 200 *Drosophila* heads per replicate using Trizol/ chloroform extraction. PolyA+ mRNA was obtained using NEBNext Poly(A) mRNA Magnetic Isolation Module. PolyA+ RNA was fragmented using the NEB Next Magnesium Fragmentation Module (NEB, E6150S) for 4 min at 95 °C for a 250 ng sample of polyA+ RNA, and RNA was repurified using the Zymo RNA clean & concentrator −5 kit (Zymo, R1013). 10% of the fragmented polyA+ RNA was saved as input control for sequencing. m⁶A-immunoprecipitation was done using the EpiMark N6-Methyladenosine Enrichment kit protocol with some minor alterations described. About 30 ul of protein G-magnetic beads (NEB, #S1430) were washed and resuspended in IP buffer (150 mM NaCl, 10 mM Tris-HCL, 0.1% NP-40). About 1.4 ul of NEB m⁶A antibody (1:178, NEB, E1610S), or 4 ul of synaptic systems antibody (1:62, Synaptic systems,202003, 2−97) was conjugated to protein G-magnetic beads (NEB, E1611A, 10015190) for 2 h at 4 °C. Beads/antibody were washed twice in IP buffer. Approximately 1 µg PolyA+ RNA was incubated with beads/antibody in IP buffer supplemented with 0.1% SUPERase-In RNase Inhibitor (Thermo Fisher; AM2696) for 2 h at 4 °C. After incubation, RNA/beads/antibody are washed twice in IP buffer, twice in low salt IP buffer (50 mM NaCl, 10 mM Tris-HCL, 0.1%

NP-40), and twice in high salt IP buffer (500 mM NaCl, 10 mM Tris-HCL, 0.1% NP-40). RNA is eluted from beads with 25 μl of RLT buffer twice, and elution was pooled and concentrated using Zymo RNA clean and concentrator kit-5 (R1015). Libraries were made using SMARTer Stranded Total RNA-Seq Kit V2 without rRNA depletion (Takara bio, 634411) for IPed and input RNA, and sequenced using Illumina HiSeq × series with 40 M paired-end reads (2 × 150bp). Library preparation and sequencing was done by Admera Health. Three biological replicates per genotype and condition were done with NEB m6A antibody, and two biological replicates were done with Synaptic Systems m6A antibody.

## M6A enrichment analysis

Regions of m6A enrichment were found for each condition using MetPeak (v.1.1)[63] with default parameters, using the input and m6A pulldown bam files as input, and with the FlyBase FB2019_05 annotation provided. Peak locations (5′ UTR, CDS, or 3′ UTR) were defined from the regions indicated by MetPeak as having significant m6A enrichment. If a peak was not contained in one region (i.e., if the peak is partly in the CDS and partly in the 3′ UTR), it was assigned to the region where more of the peak resided.

## Differential m6A peak analysis

Regions of differential methylation between two conditions (frequently called "*Mettl3*-dependent genes" or "m6A genes") were found using RADAR (v.0.2.4)[47] with input and m6A pulldown bam files as input, as well as the FlyBase FB2019_05 annotation. All replicates of SYS antibody are used for differential peak calling. The minimum cutoff for bin filtering was 15, the cutoff was set as 0.05, and the Beta_cutoff was set as 0.5. Any region with an adjusted $p$ value <0.05 was retained, and regions with a fold-change <−1 from the control (mCherry) to the knockout (*Mettl3*) at basal or heat shock (30 min) conditions were kept as dependent peaks. Non-*Mettl3*-dependent m6A were considered as all other genes expressed in the brain that did not have *Mettl3*-dependent m6A.

## M6A metaplots, heatmaps, and genome browser visualization

Heatmaps and metagene plots showing the location of m6A enrichment on a specific set of genes were constructed with using pheatmaps (v.1.0.12) and meRIPtools (v.0.2.1). Specifically, the exons of all transcripts in each gene were collapsed using the GenomicRanges (v.1.44.0) function reduce[64]. Genes with a 5′ UTR or 3′ UTR shorter than 30 base pairs, a CDS shorter than 100 bp or lacking a 5′/3′ UTR (i.e., lncRNAs) were not considered in this analysis. For each gene, the 5′ UTR and 3′ UTR were tiled in 30 evenly spaced bins, and the CDS was tiled in 100 evenly spaced bins. The number of input and m6A reads overlapping each bin was calculated and this number was divided by the bin width and library size and a normalization factor of one million to produce a normalized read per million in each bin. The heatmaps show the enrichment of m6A above input in each bin by dividing the m6A coverage by the input coverage after adding replicates from the same condition and sample type together. For heat map normalization, reads per million are normalized by the size of the bin, total reads, and library size. For genome browser snapshots, tracks visualized are log2(m6A /input) or separated input and m6A-IP tracks in supplementary figures. Tracks were made by first converting bam files to bigWig files using deepTools (v.3.5.1)[65] bamCoverage using CPM normalization, then deepTools bigwigCompare with operation log2.

## GO and pathway analysis

GO analysis for genes with non-*Mettl3*-dependent m6A or *Mettl3*-dependent m6A was conducted using FlyMine (v.53)[66]. The test correction was set to Holm−Bonferroni with a max $p$ value of 0.05. KEGG pathway analysis was done using the "enrichKEGG" function from ClusterProfiler (v.4.0.5) package in R[67]. A list of all genes with detectable expression was used as background for both GO and pathway analysis.

## Motif analysis

Motif enrichment for m6A obtained with pulldowns using either NEB or SYS antibodies was performed using HOMER (v.4.11)[68] findMotifs.pl with the parameters -rna -len 5,6,7. FASTA files containing the sequence of the RNA base pairs under each peak were compared with FASTA comprising of background sequences, which were generated by taking random regions of expressed transcripts without an m6A peak that were length-matched to the peak sequences.

## RNA-sequencing

Total RNA was extracted from brains using trizol/chloroform and Zymo RNA clean and concentrator kit-5 (R1015). The RNA-seq libraries from brains were prepared using the Tru-seq stranded mRNA library prep. Library preparation and sequencing was done by Admera Health, and sequenced using Illumina NovaSeq S4 with 40 M paired-end reads (2 × 150 bp). Three biological replicates were done for each experimental timepoint, condition, and genotype. Technical repeats and at least $n$ = 3–6 biological replicates of at least 15 brains per replicate were done as indicated in Supplementary Data 8.

## RNA-seq analysis

Raw paired-end fastqs were processed with TrimGalore (v.0.6.6) (https://github.com/FelixKrueger/TrimGalore) with default settings to remove Illumina adapters and mapped using STAR 2.7.3a[69] to the *Drosophila melanogaster* genome annotation dm6. Unmapped and improperly paired reads were filtered out of aligned bam files. Reads per gene in the FlyBase release 2019_05 were computed using an R script using GenomicRanges (v.1.44.0)[70] summarizeOverlaps that counts the number of reads overlapping with the exons of each gene in the default "union" mode. Differential expression analysis was performed using DESeq2 (v.1.32.0)[71], with count files produced by summarizeOverlaps as input. PCA plots were made using the plotPCA function in DESeq2, with variance stabilized counts as the input. MA plots were constructed from the adjusted $p$ values and baseMean values output from DESeq2, and volcano plots were constructed from adjusted $p$ values and fold changes reported by DESeq2. Normalized counts produced by DESeq2 were used to show expression levels. Differentially expressed genes were considered to be any gene with $p$-adjusted values of <0.05.

## Actinomycin D assay

Brains from $w^{1118}$ (BL5905), mCherry RNAi, and *Mettl3* RNAi adults were dissected and incubated in 250 ul of Schneider's *Drosophila* Medium that was pre-warmed to 25 °C in RNase-free 1.5 ml microfuge tubes. Fifteen to 20 brains were dissected for each timepoint. About 169.5 ug/ml of actinomycin D (Sigma-Aldrich, A1410) was added to each sample. Samples were incubated at 25 °C, shaking at 300 rpm for indicated timepoint, and mixed by pipetting every 30 min. After the incubation time, samples were collected, spun at room temp, washed in cold RNase-free PBS, and processed in Trizol. Brain total RNA was collected using trizol/chloroform and Zymo RNA clean and concentrator kit-5 (R1015). RNA was then used for RT-qPCR. About 400 ng of RNA is used per cDNA reaction. cDNA is also quantified by Qubit ssDNA Assay (Invitrogen, Q10212) before use in RT-qPCR experiments.

## Puromycin assay

Food was made by mixing 600 μM puromycin (Sigma-Aldrich, P8833-100MG) with 2% agar + 5% sucrose. Adult flies were placed onto puromycin food for 24 h. Brains were dissected from flies after 24 h of feeding and samples were processed for western blot analysis. Protein loading was determined by the Ponceau S stain of the membrane.

## Quantification and statistical analysis

Statistical tests used were performed on GraphPad Prism (v.9) and are indicated in the figure legend. *P* values of <0.05 were considered significant. Unpaired two-tailed *t*-tests were used when comparing two groups; One-way ANOVA was used when comparing multiple groups, followed by Tukey's post-test when each group was compared against every other group, Sidak's post-test when pre-defined groups were compared to each other, or Dunnett's test when compared to a defined control sample. Two-way ANOVA was used when there were two factors in the analysis (usually RNA amount and HS condition). Fisher's exact test was used when comparing m$^6$A vs non-m$^6$A genes (Fig. 4). One-sided Hypergeometric test was used to compare Venn diagram overlaps.

## Reporting summary

Further information on research design is available in the Nature Research Reporting Summary linked to this article.

## Data availability

The Raw sequencing data generated in this study have been deposited in the Gene Expression Omnibus under accession code GSE178955. Full motif analysis are available in Supplementary Data 1. The m$^6$A peak RADAR data used in this study are available in Supplementary Data 2. Go term and Kegg pathway analysis are available in Supplementary Data 3. The differential gene expression tables from RNA-seq analysis generated in this study are provided in Supplementary Data 4 and 5. HS gene lists are provided in Supplementary Data 6. Fly genotypes and primers used are provided in Supplementary Data 7. Mapping rates for all sequencing experiments are provided in Supplementary Data 8. Any additional inquiries can be directed to the corresponding author. Source data are provided with this paper.

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

## Acknowledgements

We thank China N. Byrns and Ananth R. Srinivasan for critical manuscript feedback; M. Kayser, K. Jordan-Sciutto, G. Ming, and B. Gregory for valuable input. We thank the *Drosophila* Genomics Resource Center, supported by NIH grant 2P40OD010949, and the Bloomington *Drosophila* Stock Center (NIH P40OD018537) for fly lines. We thank the Transgenic RNAi Project (TRiP) at Harvard Medical School (NIH/NIGMS R01-GM084947) for developing transgenic RNAi fly stocks used in this study. This work was supported by T32-GM007229 and F31-AG063470 (to A.E.P.), R35-GM133721 (to K.F.L.), and R35-NS097275 (to N.M.B.).

## Author contributions

A.E.P. conceived, designed and performed experiments, statistical analysis, bioinformatic analysis, and analyzed data. E.J.S. designed and performed bioinformatic analysis. H.S. performed mass spec experiments. K.F.L. supervised research and analyzed data. N.M.B. conceived, designed experiments, analyzed data, and supervised the research. E.J.S. and K.F.L. gave input and edited the manuscript. A.E.P. and N.M.B. wrote the manuscript.

## Competing interests

The authors declare no competing interests.
