## [Peer Review File · Nature Communications]

Mettl3-dependent m6A modification attenuates the brain stress response in *Drosophila*Reviewers' comments:

Reviewer #1 (Remarks to the Author):

In this study Perlegos et al presents the role of m6A during the stress response in *Drosophila*. The authors showed that m6A level is specifically elevated in brain mRNA after heat shock and that, surprisingly, m6A has a negative influence on stress resistance. Indeed the survival rate was higher after HS when Mettl3 was depleted ubiquitously or only in neurons or glia. M6A is enriched in the 5'UTR of fly heads mRNA and is also found along the CDS. The former peaks are Mettl3 dependent while the latter ones are independent. Mettl3 restricts the expression of heat shock protein genes as well as more generally the transcript and protein levels in the brain of Mettl3 dependent methylated genes. Lastly the authors showed that the readers Ythdf and Fmr1 also restrict gene expression in the brain but in contrast to Mettl3 are not involved in stress resilience.

This study is interesting as it uncovers a new facet of m6A during stress conditions. In contrast to its expected function m6A renders the fly brain more susceptible to stress. Its selective role in the brain is quite striking. The overall findings should be of broad interest for readers but additional controls are needed to validate the major conclusions.

Major comments

1- The Mettl3 independent peaks are surprising given that other studies have shown that the knock out of Mettl3 almost completely abolished m6A in mRNA. Can the authors formally exclude that these peaks are due to non-specific recognition by the antibodies? Recognition by two antibodies raised against the same antigen is not a very solid demonstration. Could some of these peaks be validated with an orthogonal approach? The de novo motif analysis identified the m6A consensus site but only at the 6th or 7th position. If taking only the Mettl3-dependent peaks does the consensus site become the first motif identified? And what is the consensus site for the Mettl3-independent peaks?

2- Most of the experiments were performed using RNA interference against Mettl3. Since off target effects can occur the authors should confirm their findings using Mettl3 mutant alleles as well as Mettl14 alleles.

3- Currently it is not clear how Mettl3 renders the flies more susceptible to stress. The main identified readers in *Drosophila* are the two YTH proteins. Can the authors test the potential involvement of the second YTH protein in stress resilience?

Minor comments

1- Fig1a: the authors should increase the number of n by at least one, for the experiment and statistical analysis. It is not very common practice to perform statistical analysis on n=2 for biochemical experiments.

2- Fig1b: The extent/fold change of m6A increase at 30'HS as measured by LCMS/MS is substantially weaker as that measured by dot blot. Any particular reason for this? For better comparison, all the time points of HS should be measured by LCMS/MS as it was done by dot blot.

3- Fig1c: Increase the n number to complete an experiment, even if additional replicates are present in supplementary data. More importantly, the two membranes do not seem to align.

4- Is there a particular reason of using only male animals?

5- Fig2a: There is a discrepancy between the age of flies mentioned in the schematic and the legend. Why was the time point of HS changed from 30' to 90'?

6- Fig2c: There is an inconsistency in the HS duration (60' here)

7- To firmly conclude that Mettl3 protein level is not affected upon HS it would be useful to do a

protein profile of Mettl3 by WB at the different time points of HS as measured in fig1 for m6A.

8- Fig3a: There is an increased occurrence of m6A peaks in the CDS of Mettl3-RNAi, basal as well as HS. Could the authors comment on/explain this? Is it a technical or biological phenomenon?

9- The heat map for Hsp70 Aa/Ab in figure 3c does not match with figure 3e peaks. Is it due to a normalization issue?

10- 81% of all expressed genes lack m6A. Since there are 4401 genes that contain m6A does this imply that more than 23000 genes are expressed in the brain?

11- Figure 6b (gray shaded) 5` UTR is mislabeled as 3` UTR.

12- In the discussion the authors claimed that there is a modest overlap of Ythdf and Fmr1 targets with Mettl3 dependent targets. This is unexpected as previous works have shown much larger overlaps. Since the authors compare targets from larval brains and S2 cells with adult heads this could easily explain the modest overlap.

Reviewer #2 (Remarks to the Author):

This manuscript by Perlegos et al reports a very interesting phenomenon that Mettl3-siRNA *Drosophila* have a better chance of survival after heat shock (HS) insult. The study shows that knocking down Mettl3 specifically in neurons or astrocytes both improve survival rates, thus clearly demonstrating that this phenomenon involves Mettl3 function in the nervous system. The authors then carried out two candidate searches to examine the underlying mechanisms: 1. to focus on heat shock chaperon proteins and examine whether their adaptive expression may be changed upon Mettl3-siRNA; 2. identify m6A modified genes in brain/head both at basal and HS conditions and examine the expression dynamics of three groups (no-m6A, constant, and Mettl3-dependent m6A). In general, the data suggested more differential transcripts with Mettl3-dependent m6A sites with a bias to upregulation. The genome-wide approach further suggested specific gene pathways such as MAPK signaling and neurogenesis that contain Mettl3-dependent sites and are upregulated in response to HS. These processes may be particularly important for explaining the survival rate improvement in Mettl3-siRNA as they may prepare the animals for the HS beforehand. To evaluate global translation, the authors used puromycylation assay and showed an increase in the brain upon Mettl3-, Ythdf-, and Fmr1-siRNA treatments. In addition, the authors also showed that overexpression of Mettl3 had opposite effect on expression of the tested targets. In general, a role of Mettl3 and m6A as repressing expression of specific mRNAs specifically in the nervous tissue, both at basal state and under heat shock conditions, has revealed. In terms of stress response, the authors suggested a "dampening" role of Mettl3-dependent m6A on brain's transcriptional and biological response to stress, which may be related to the vulnerability of the brain to neurodegenerative diseases.

The approaches in this paper are valid the results are VERY interesting and show consistency with previous published work. Given the complexity of m6A regulation and the wide impact on gene expression, I do suggest some more experiments to be conducted and clarification in the paper before authors' conclusions can be comfortably accepted for publication.

1. Is the improved survival rate due to changes in m6A (the authors showed temporary increase of m6A in poly(A) after HS) ? Although Mettl3-siRNA clearly shows better survival rates, it is not clear whether this is due to m6A-dependent mechanisms. Given that m6A-independent role of Mettl3 has been reported, another perturbation experiment targeting m6A is needed. Especially the authors show that Ythdf- and Fmr1-siRNA did not improve survival rate. Although the authors seem to have validated Mettl14-siRNA in supplementary figures, I could not find HS-related phenotypes in the knock-down animals (I may have overlooked). If the authors have not shown it, it will be important to include Mettl14 knockdown effect in survival assay. Alternatively, targeting other methyltransferase complex to disrupt methylation activity can be used to serve the same purpose.

2. The authors proposed Mettl3-mediated dampening effect may explain the brain-specific responses to stress. In Supplementary Figure S2a, the data showed that brain has “delayed” and “less dramatic” changes in heat shock chaperon proteins. This is very interesting, but how is the dynamics affected in Mettl3-siRNA animals? Only one time-point of 30min after HS was presented at the protein level. At RNA level, it will be interesting to extract the data for heat shock chaperon genes as an independent set. In addition, given the known uncoupling between transcription and translation of Hsp70, it will be necessary to know Mettl3-siRNA effect on the heat shock protein responses at protein levels.

3. The differential expression results in the three groups (no-m6A, constant, and Mettl3-dependent m6A) are very interesting. However, the dynamic changes of methylation status at individual gene level was not sufficiently addressed. The individual IGV examples shown by the authors indicated loss of 5'UTR peaks and in the case of DnaJ-1, also loss in CDS (Fig 3 and 4), thus these individual examples can not explain the Mettl3-dependent increase m6A level shown in dot-blot in Fig 1. Some individual genes (possibly abundant ones) must have increased m6A modification to explain the results in Fig 1 and individual examples of those should be given.

4. Puromycylation experiments were only conducted in basal state. Although global protein expression is known to be reduced during HS, which makes labeling efficiency lower thus detection harder, it is important to know the effect of Mettl3-siRNA during HS response. For example, 1.5 hrs HS + recovery condition can be used to allow sufficient labeling.

Minor points:

1. Fig 2f. May the labels underneath the blot be UAS-Mettl3?
2. Fig 4f. IGV view of fl(2)d show lots of reads in introns, possibly misalignment?
3. Fig 4i. It seems that dtau protein WB data under hs 30 min condition is missing.

Reviewer #3 (Remarks to the Author):

In this manuscript, the authors investigate how m6A contributes to the brain's stress response in flies by using acute heat shock stress (HS). They find that m6A levels are elevated in the brain in flies and that they further increase in the adult brain after HS. Furthermore, they identify a subset of RNAs that are Mettl3 methyltransferase targets in the brain with 5'UTR m6A and show that a small number of these transcripts undergo dynamic m6A regulation after HS. Furthermore, they show that Mettl3 depletion in flies leads to increased survival after HS. The authors use correlative analyses, global translation assays, and examination of a few specific m6A-containing RNAs to develop a model in which HS causes changes to m6A in some RNAs, and that loss of Mettl3 contributes to HS resilience by promoting the increased stability and increased translation of 5'UTR m6A targets. Although this is an interesting model, unfortunately the experiments do not support the broad conclusions in the manuscript. Many of the analyses remain correlative and are at times unclear or difficult to interpret. In my opinion, there are substantial experiments that would need to be done to support the authors' conclusions. I have discussed some of my concerns below and hope that the authors will find this helpful.

Major Comments

1. The authors only show a small number of RNAs that have Mettl3-dependent changes in m6A, which suggests that this is not a global m6A-dependent stress response but rather a handful of RNAs with differential methylation. I also have concerns about the Merip seq data; it does not seem to be rigorously analyzed (see below) and the vast majority of sites seem to be present with Mettl3 reduction. Are these peaks due to noise in the MeRIP analysis or to some new set of m6A sites/methylated RNAs for which m6A is deposited by a different MTase other than Mettl3? The

authors state in the discussion that some MTase components direct m6A to distinct regions. While there is some evidence for this, in the end all Mtase components rely on Mettl3. Indeed, the mass spec data shows a very robust depletion of m6A in Mettl3 RNAi brains (Fig 1), suggesting that Mettl3 does in fact direct the methylation of most m6A sites and therefore leading to the assumption that this is due to noise in the MeRIP data.

2. It is unclear from the methods section how m6A peaks were called among replicates. Were only the peaks common to all biological replicates used? What is the variability in peaks between each replicate? These steps are critical for ensuring the robustness of the MeRIP approach, as previous studies have shown that stochastic peak calling is a common feature of MeRIP datasets. Figure S3b shows high overlap between antibodies at the gene level, but overlap between individual replicates and at the peak level is not shown. Furthermore, I could not find a list of m6A peaks in their data files.

3. It is unclear that the IP/input calculation properly accounts for read coverage in each MeRIP library. MeRIP is notorious for stochastic peak calling due to variability in IP efficiency. The gene tracks in figure 4f, for instance, show a lot of Mettl3-independent peaks and even for some the 5'utr mettl3-dependent peaks have a high amount of reads in Mettl3 depleted conditions. It would be more useful to show these gene tracks as separate tracks with IP and input reads on the same track so that we can see how much the IP is enriched relative to input. Also, how exactly was read depth taken into account when calculating IP/input? RADAR has methods for accounting for this but these were not mentioned in the Methods section. Also, the methods section says that a β_cutoff (log fold change cutoff) of 0.5 was used, but then later it says that regions with a fold change <-1 were considered dependent peaks. It is not clear to me based on this description what exactly the thresholds were for fold-change determination.

4. It is also not clear why the basal RRACH m6A consensus is not enriched in Merip peaks under basal conditions for either antibody. The authors claim that their motif analysis uncovered similar peaks to those found by Lence et al in S2R+ cells, but this does not seem to be the case under basal conditions. They do see RAC motifs in HS conditions, but curiously they have reported the 7th and 6th ranked motifs. Why? What were the top motifs?

5. In figure 1a, the increase in m6A via dot blot seems to track with the amount of RNA loaded. In theory, the relative intensity should not change with RNA amount, but the quantification indicates otherwise. Why is this? There are also wide variabilities in the intensity increase in m6A after HS across dot blots (eg, compare fig. 1a to 1c). I suggest including more replicates, especially since the MS data in panel 1b show a much less dramatic increase in m6A compared to the dot blots. In general, there is an over-reliance on dot blots throughout to confirm loss of m6A. Dot blots are a poor measure of m6A quantification and suffer from non-specificity issues related to recognition of non-m6A modifications by the antibodies. MS should be done, in particular to confirm the loss of m6A in Mettl3 RNAi brain tissue, not merely whole heads.

6. It is an overstatement to say that the brain has a dampened stress response based on the results of Figure S2, especially since the levels of Hsp70 are quite strongly induced in the brain after HS.

7. Figure 2e indicates a relatively modest effect of Mettl3 RNAi on Hsp70 levels in the brain, and the western blot shown in the figure clearly has an increase in both tubulin and Hsp70. There is an over-reliance on western blot densitometry for quantification of protein amounts throughout the manuscript. This is concerning for some antibodies which have questionable specificity (eg, futsch, also dtau which appears to have two bands, one of which is cut off from the blot shown in Figure 4). A more robust readout for translation such as polysome analysis or quantitative western blot would help to confirm these results.

8. The authors classify RNAs as no m6A, constant m6A, or Mettl3 dependent m6A. It is unclear what the "constant" category is offering. Clearly the peaks in these RNAs are not Mettl3-dependent m6A peaks...so what do the authors think they are? To me it suggests very high background in their MeRIP-seq data (see comments above), which means that these constant RNAs are no different than non-methylated RNAs. Also, few genes show HS-dependent changes so it is unclear

how the authors anticipate m6A altering so many transcripts when the methylation levels themselves do not change.

9. It is not clear what the minimum fold change was for RNA-seq analysis in terms of identifying up/downregulated genes. Based on the mean log₂FC listed in the text, as well as their gene expression plots for different classes of RNA, it appears that a lot of these transcripts are not drastically changed.

10. The authors show data that does not quite fit how they phrase the analysis. Line 358 says, "more Mettl3-dependent m6A genes were differentially expressed both with HS and with Mettl3 RNAi (18.2% of the DE genes in HS) compared to constant genes (6.5%) or no-m6A genes (6.3%)". These numbers are percentages of the HS vs Cntrl DE genes, but if you calculate the percentage instead relative to Mettl3 RNAi vs Cntrl DE genes, you get much more similar numbers across all groups (50-60%).

11. The data shown in Fig 5e,f (differential expression of genes in different GO/KEGG categories) includes for the most part a very small number of genes. Therefore these analyses should not be broadly interpreted to support more general claims of global effects of Mettl3 and HS on RNA abundance and stress resilience.

12. The data linking m6A to Ythdf and Fmr1 are mostly correlative; while the authors show interesting effects of Ythdf and Fmr1 depletion on global protein synthesis, we do not know how many of these proteins are encoded by RNAs with m6A. This is important to know in order to fit the model that these proteins are "key regulators that promote Mettl3-dependent m6A inhibition of protein translation in the brain" as the authors conclude (lines 414-417). Also, since Fmr1 is known to inhibit translation through m6A-independent mechanisms, it is very likely that the effects they observe with Fmr1 depletion are occurring independently of the m6A pathway.

13. The broad changes in protein synthesis after Mettl3 depletion are inconsistent with the relatively few transcripts that show Mettl3-dependent m6A in the brain. Furthermore, the Sunset assays were not performed in HS brains, which would speak better to the potential role of Mettl3 in regulating translational responses to stress.

14. The Sunset assays (Fig 6) have large variability in the control conditions across samples which is not reflected in the densitometry analysis. What is the source of this variability?

15. The data showing the effect/lack of effect of Ythdf1 and Fmr1 on stress sensitivity should be shown. It would also be good to include the effect of Mettl3 overexpression on stress resilience.

Minor Comments

It is not clear to me how the heat shock was performed. The Methods section describes a stress sensitivity assay with HS for 1.5h at 38.5C, but other experiments suggest 30min at 38.5C. The authors should specify when different conditions for the HS were used across experiments and why.

The callouts to Figs 4f-I in the text are inaccurate.

Fig 6b: 5'UTR mislabeled as 3'UTR

It seems inaccurate to refer to head m6A peak RNAs as brain targets. I recognize that m6A is elevated in the brain relative to whole head/whole fly, but unless the authors show that head minus brain lacks m6A, or unless they perform MeRIP on brain specifically, they cannot know for sure whether these peaks come from the brain.

Response to the Reviewer's concerns:

We thank the Editor and Reviewers for their constructive comments on our work. The main concerns of the Reviewers were a lack of clarity in the presentation of the m⁶A peaks and the highlighting of *Mettl3*-independent m⁶A peaks; lack of confirmation of *Mettl3* RNAi effects with additional *Mettl3* mutants; over-reliance on dot blots vs mass spec for m⁶A levels; and the suggestion to implicate the m⁶A YTH proteins in the brain stress response.

We have now addressed these concerns in full with compelling new data and have substantially revised the manuscript.

- We now show that the critical findings are confirmed with *Mettl3* catalytic and null mutants and *Mettl14* RNAi, confirming that the stress effects seen are due to the catalytic m⁶A activity of *Mettl3* (Fig. 2d, Fig. S3a-b).
- We have now added additional mass spectrometry analysis of m⁶A on polyA⁺ RNA that supports our findings (Fig. 1).
- We now show extensive data on the replicates for the m⁶A-seq data and revised our discussion of the *Mettl3*-independent peaks in the m⁶A-IP. We note that our data agree with other published data on *Mettl3*-dependent peaks being largely in the 5'UTR, and that others see *Mettl3*-independent peaks^{1,2}. However, we have now thoroughly revised the manuscript to focus on solely *Mettl3*-dependent m⁶A peak genes and their role in modulating the stress response.
- We performed new experiments to gain insight into the role of the YTH canonical m⁶A reader proteins in the *Mettl3*-m⁶A-heat shock resilience pathway. Upon examining *Ythdc1* and *Ythdf*, our findings indicate the critical features of *Mettl3* loss are recapitulated by *Ythdc1* (Fig. 7).

We highlight that the addition of extensive new data has confirmed and strengthened our original findings. We have spent considerable time re-writing the manuscript so that it reads more smoothly and clearly.

Detailed listing of the revisions:

New data analyses, text, figures, tables and methods:

1. Figures Main

- Fig 1: Additional replicate of heat shock (HS) time course dot blot for n=3 and statistics added. Additional LC-MS with 24 recovery post heat shock and validation of *Mettl3* knockdown m⁶A levels in the brain.
- Fig 2: Additional heat stress sensitivity assays using *Mettl3* catalytic mutants, indicating dependence of the response on the catalytic domain.
- Fig 3: Changed metagene frequency plot to a normalized read coverage plot of m⁶A-IP/ input of *Mettl3*-dependent m⁶A genes that gives a more accurate visual representation of m⁶A enrichment sites without relying on peak calling.
- Fig 4: Removed analysis of *Mettl3*-independent peak to focus on robust *Mettl3*-dependent m⁶A genes and their response to HS and to *Mettl3* knockdown. Additional analysis of *Mettl3*-dependent vs non *Mettl3*-dependent gene sets to account for statistics in upregulation skew observed with HS and *Mettl3* RNAi.
- Fig 5: Highlighted in red regions of heat map that show statistically significant m⁶A sites upon heat stress in the 5'UTR. Included additional western protein blots with

- heat stress time course of Hsp70, DnaJ-1, and stv protein levels in control and *Mettl3* knockdown brains.
- Fig 6: Additional puromycin assay with heat stress, as well as puromycin assay with *Ythdc1* knockdown and upregulation showing consistent results as *Mettl3* knockdown.
 - Fig 7: Addition of heat stress sensitivity assays with *Ythdc1* perturbation. New brain RNA-sequencing with *Ythdc1* RNAi, and protein blots of HS chaperone Hsp70 levels in basal and HS conditions with *Ythdc1* knockdown.
 - Figure 8 (Model): Updated model reflecting additional data with reader protein *Ythdc1* and focused model on *Mettl3*-dependent m6A genes.
 - Fig S3: Addition of *Mettl14* knockdown and *Mettl3* catalytic mutant sensitivity assay showing increase heat stress resilience. Addition of heat stress time course western blot of *Mettl3* protein levels in control brains showing no change in *Mettl3* level with HS.
 - Fig S4: Additional motif analysis with *Mettl3*-dependent m⁶A sites. All motif files are now included in Supplementary Data 1. Comparison of m6A/input signal between replicates in control and HS conditions for control and *Mettl3* flies. Our motifs agree with published fly data^{3,2,1}.
 - Fig S5: fl(2)d and futsch, western protein blots were moved to supplement, and included genome browser plots for visualization of *Mettl3*-dependent sites in 5'UTR. Genome browser plots show input and IP separate, not log(m⁶A/input). Addition of *Mettl3*-dependent gene Draper western blot in basal and HS with *Mettl3* RNAi.
 - Fig S7: Highlighted in red regions of heat map that show statistically significant m⁶A sites in basal conditions on heat shock genes in the 5'UTR. Genome browser plots for visualization of *Mettl3*-dependent sites in 5'UTR on *Hsp70*, *DnaJ-1*, and *stv*. Genome browser plots show input and IP separate instead of log(m6A/input). Additional western blots of Hsp70 protein levels at baseline in control and *Mettl3* knockdown brains show increase of protein levels with *Mettl3* knockdown. Additional quantitative western blot of Hsp70 protein levels with HS and serial dilutions of brain protein show increased levels of protein in *Mettl3* knockdown brains.
 - Fig S9: Additional Puromycin assay showing increased puromycin incorporation with *Mettl3* catalytic mutant, *Mettl3* null mutant, and *Mettl14* knockdown animals. Additional puromycin assay with puromycin plus actinomycin D (transcription inhibitor) in *Mettl3* knockdown animals and shows increased puromycin incorporation.
- 2. Text has been updated with new results and data analyses throughout the manuscript**
- 3. Files:**
- Data S1: Added full motif analysis output
 - Data S2: Added supplementary data of RADAR differential methylation output files.
 - Data S3: Added additional Go term analysis from data of RNA-seq in *Ythdc1* knockdown brains in basal and heat stress conditions.
 - Data S5: Added additional differential expression files of RNA-seq in *Ythdc1* knockdown brains in basal and heat stress conditions.
- 4. Additional methods:**
- Additional explanation of heat stress sensitivity assay for use of mild HS or longer HS for survival assays.

- M⁶A-IP seq analysis updated to be transparent how peak calling program accounts for replicates, clarify threshold of¹⁻⁴ RADAR peak calling to determine *Mettl3*-dependent peaks, and explanation how read depth taken into account when calculating IP/input.
- Additional methods added to clarify why male flies are used.

By performing these data analyses and adding new data we have strengthened the significance of the results and —importantly— these new data have robustly reinforced and clarified the main findings of the original data set.

These findings include:

1. *Mettl3* loss of function promotes heat stress survival, indicating m⁶A can impede the brain stress response, validated using additional catalytic *Mettl3* mutants.
2. *Mettl3*-dependent m⁶A genes are a brain-enriched class of genes that becomes upregulated upon heat stress and upon *Mettl3* knockdown.
3. *Mettl3* knockdown increases heat stress chaperone RNA and protein levels in the brain and increased global protein translation.
4. A novel *ex vivo* brain actinomycin assay showing that *Mettl3* knockdown influences RNA decay of critical heat stress chaperones.
5. *Mettl3* knockdown decreases levels of nuclear reader protein *Ythdc1*. *Ythdc1* knockdown allows for stress resilience, similar to *Mettl3* knockdown, and increases levels of heat stress chaperones.
6. This study is the first m⁶A-IP sequencing with heat stress *in vivo*, by utilizing *Drosophila* heads with *Mettl3* knockdown; brain tissue RNA-sequencing with *Mettl3* knockdown and nuclear reader protein *Ythdc1* knockdown, with heat stress and stress recovery time-course.

Point by point Response:

We include the Reviewers comments in black and our response in blue.

Major comments

1- The *Mettl3* independent peaks are surprising given that other studies have shown that the knockout of *Mettl3* almost completely abolished m⁶A in mRNA. Can the authors formally exclude that these peaks are due to non-specific recognition by the antibodies? Recognition by two antibodies raised against the same antigen is not a very solid demonstration. Could some of these peaks be validated with an orthogonal approach? The *de novo* motif analysis identified the m⁶A consensus site but only at the 6th or 7th position. If taking only the *Mettl3*-dependent peaks does the consensus site become the first motif identified? And what is the consensus site for the *Mettl3*-independent peaks?

Thank you for your comment. Although these peaks are consistent between antibodies we appreciate that we cannot formally exclude that these peaks are due to non-specific recognition. At this time, we are constrained by our technique of m⁶A-IP and m⁶A-Clip-seq, and are not currently able to expand to nanopore or antibody independent DART-seq technologies for use in *Drosophila* brains. The m⁶A field recognizes that it is extremely difficult to ablate all m⁶A and currently struggles with this observation of *Mettl3*-independent m⁶A peaks. Many reports and datasets observe *Mettl3*-independent m⁶A sites; for example, recent independent work in *Drosophila* confirms the presence of these *Mettl3*-independent peaks^{1,2}. Due to the considerable amount of m⁶A that persists after *METTTL3* knockout,⁴⁻⁸ it is speculated that other genes or isoforms⁹ may have a role in *Mettl3*-independent m⁶A deposition on mRNA^{7,10-13}. When we sort

out dependent and independent peaks we see that the consensus motif AAAC comes out in 3rd and 4th positions respectively in *de novo* motif analysis (Sup. Fig. 4e and Sup. Data 1).

With the Reviewers' comments in mind, we have now focused the manuscript on the genes with *Mettl3*-dependent m⁶A peaks. Given the unresolved nature of the *Mettl3*-independent m⁶A (which we see here with knockdown of *Mettl3* in head tissue, and in Wang et. al.¹ and Kan et. al.² with head and whole flies, respectively, in *Mettl3*-null mutation situations), we include these genes in the category of non-*Mettl3*-dependent m⁶A.

2- Most of the experiments were performed using RNA interference against *Mettl3*. Since off target effects can occur the authors should confirm their findings using *Mettl3* mutant alleles as well as *Mettl14* alleles.

We have now confirmed critical findings with null and catalytic mutants of *Mettl3*. Specifically, the catalytic *Mettl3* mutant (Catalytic Mutant is studied as catalytic/deficiency) shows an increase in HS stress resilience (Fig. 2d), and both catalytic mutant and null mutants have increased levels of puromycin compared to control brains (Sup. Fig. 9), indicating that the increased puromycin and stress resilience are associated with the catalytic activity of *Mettl3*. We have also confirmed critical results with *Mettl14* RNAi in that *Mettl14* reduction also confers heat stress resilience (Sup. Fig. 3a) and increased puromycin incorporation in brains compared to control (Sup. Fig. 9c).

3- Currently it is not clear how *Mettl3* renders the flies more susceptible to stress. The main identified readers in *Drosophila* are the two YTH proteins. Can the authors test the potential involvement of the second YTH protein in stress resilience?

Thank you for this suggestion. We now show that *Ythdc1* confers the key features of stress of *Mettl3* (Fig.7). Specifically, our brain RNAseq shows that *Ythdc1* levels are reduced upon *Mettl3* knockdown (Fig. 7a), and *Ythdc1* RNAi upregulates heat stress chaperone levels (Fig. 7d-f). *Ythdc1* knockdown animals are also heat stress resilience (Fig. 7b). Thus, we suggest that *Ythdc1* is a key m⁶A reader protein in conferring heat stress resilience.

We have also included more data to address the basal levels of heat stress chaperones. *Mettl3* RNAi shows higher levels of Hsp70 both basally and with heat stress. At basal conditions, we suggest this is akin to stress pre-conditioning. Thus, we suggest that higher levels of central stress protecting proteins confer the heat resilience with knockdown, and a greater heat sensitivity effect with normal *Mettl3* gene function in the brain.

Minor comments

1- Fig1a: the authors should increase the number of n by at least one, for the experiment and statistical analysis. It is not very common practice to perform statistical analysis on n=2 for biochemical experiments.

We have increased the N for the dot blot experiment (Fig. 1a) to n=3 and reanalyzed statistical data.

2- Fig1b: The extent/fold change of m6A increase at 30'HS as measured by LCMS/MS is substantially weaker as that measured by dot blot. Any particular reason for this? For better comparison, all the time points of HS should be measured by LCMS/MS as it was done by dot blot.

Thank you for this comment. We have now performed additional LCMS/MS assays (24 h recovery time-point) (Fig. 1b). We did not test additional HS time points since we focus on the 30 min HS time point for sequencing. We also performed additional LCMS/MS of brain *Mettl3* knockdown, which shows decreased m⁶A levels (Fig 1c).

We note that, from DNA m⁵C dot blot assays, the field acknowledges that the intensity of immunoblotting is not linear like LC-MS/MS quantification. Thus, it is recognized that immunoblotting often enlarges differences detected. To focus our analysis, we removed most dot blots from the main figures and performed additional LCMS/MS assays as noted above.

3- Fig1c: Increase the n number to complete an experiment, even if additional replicates are present in supplementary data. More importantly, the two membranes do not seem to align.

Thank you for your comment. We have removed original Fig. 1c from the main figure as the more stringent LC-MS/MS data are now in Fig. 1b.

4- Is there a particular reason of using only male animals?

We typically use male flies for consistency in the experiments, to avoid issues in lifespans due to egg laying. We have now added a sentence to the methods to make this clear.

5- Fig2a: There is a discrepancy between the age of flies mentioned in the schematic and the legend. Why was the time point of HS changed from 30' to 90'?

Thank you highlighting this. This was a typo and the schematic has been fixed to 6 d. For heat shock survival, we follow the standards in the field. The standard is to use a longer severe heat shock of 1.5 h when assessing survival of flies post HS stress (see refs ^{14,15}). The standard to examine the molecular basis of the sensitivity to HS stress in *Drosophila* is 30 min. We consistently use a short heat shock as a mild stressor and a longer heat shock for a severe stressor to assess stress survival sensitivity. We have added this additional clarification in the methods.

6- Fig2c: There is an inconsistency in the HS duration (60' here)

We apologize for the lack of clarity. As the *UAS-Mettl3* flies are severely stress sensitive, we needed to reduce the stress time to 1 h to have some (at least 5-10%) of the flies surviving. We have now made this clear in the figure legend (Fig. 2).

7- To firmly conclude that *Mettl3* protein level is not affected upon HS it would be useful to do a protein profile of *Mettl3* by WB at the different time points of HS as measured in fig1 for m⁶A.

We have now examined protein changes in *Mettl3* by western immunoblot with heat stress and stress recovery in the brain: there are no significant changes with HS or recovery (Sup. Fig. 3c).

8- Fig3a: There is an increased occurrence of m⁶A peaks in the CDS of *Mettl3*-RNAi, basal as well as HS. Could the authors comment on/explain this? Is it a technical or biological phenomenon?

Thank you for this comment. Our original method of plotting the peak density throughout the gene body showed the *relative* occurrence of peaks, making it unclear whether the apparent

increase in CDS peaks is due to increased signal at the CDS, or simply that the loss in 5' UTR peaks led to a distribution shifted more towards the CDS. We have replaced the figure with one that shows the normalized coverage (m^6A/input) signal across the gene body, which shows more accurately the signal prior to peak calling. Fig. 3a shows normalized coverage from all *Mettl3*-dependent m6A genes, and we observe a very slight non-significant increase in CDS upon *Mettl3* RNAi likely driven by a few genes (seen RADAR peak calling Sup. Data 2). Overall the average LogFC peak change in CDS was -0.11.

9- The heat map for Hsp70 Aa/Ab in figure 3c does not match with figure 3e peaks. Is it due to a normalization issue?

The heat map in Fig. 5a shows in HS conditions the m^6A divided by input reads. There are red m^6A enrichment regions across CDS of *Hsp70Aa/Ab* in the heat map which is shown in the genome browser of *Hsp70* in Fig 5b. In basal conditions (shown in Sup. Fig. 7a-c), there is m^6A enrichment in the 5'UTR that is not present in HS conditions.

10- 81% of all expressed genes lack m6A. Since there are 4401 genes that contain m6A does this imply that more than 23000 genes are expressed in the brain?

This has been corrected. The correct % of *Mettl3*-dependent genes is 14%. The number of total genes we detect in the brain from RNA-seq and IP-seq is about 14,943. Number of *Mettl3*-dependent genes is 2120.

11- Figure 6b (gray shaded) 5' UTR is mislabeled as 3' UTR.

Thank you for pointing this out. In the restructured manuscript, this figure has been removed.

12- In the discussion the authors claimed that there is a modest overlap of Ythdf and Fmr1 targets with *Mettl3* dependent targets. This is unexpected as previous works have shown much larger overlaps. Since the authors compare targets from larval brains and S2 cells with adult heads this could easily explain the modest overlap.

Thank you for this comment. This is likely the reason of the modest overlap. We have now removed this analysis to focus on the Ythdc1 m^6A reader protein.

Reviewer #2 (Remarks to the Author):

This manuscript by Perlegos et al reports a very interesting phenomenon that *Mettl3*-siRNA *Drosophila* have a better chance of survival after heat shock (HS) insult. The study shows that knocking down *Mettl3* specifically in neurons or astrocytes both improve survival rates, thus clearly demonstrating that this phenomenon involves *Mettl3* function in the nervous system. The authors then carried out two candidates searches to examine the underlying mechanisms: 1. To focus on heat shock chaperon proteins and examine whether their adaptive expression may be changed upon *Mettl3*-siRNA; 2. Identify m6A modified genes in brain/head both at basal and HS conditions and examine the expression dynamics of three groups (no-m6A, constant, and *Mettl3*-dependent m6A). In general, the data suggested more differential transcripts with *Mettl3*-dependent m6A sites with a bias to upregulation. The genome-wide approach further suggested specific gene pathways such as MAPK signaling and neurogenesis that contain *Mettl3*-dependent sites and are upregulated in response to HS. These processes may be particularly important for explaining the survival rate improvement in *Mettl3*-siRNA as they may prepare the animals for the HS beforehand. To evaluate global translation, the authors used puromycylation

assay and showed an increase in the brain upon *Mettl3*-, *Ythdf*-, and *Fmr1*-siRNA treatments. In addition, the authors also showed that overexpression of *Mettl3* had opposite effect on expression of the tested targets. In general, a role of *Mettl3* and m6A as repressing expression of specific mRNAs specifically in the nervous tissue, both at basal state and under heat shock conditions, has revealed. In terms of stress response, the authors suggested a “dampening” role of *Mettl3*-dependent m6A on brain’s transcriptional and biological response to stress, which may be related to the vulnerability of the brain to neurodegenerative diseases.

The approaches in this paper are valid the results are VERY interesting and show consistency with previous published work. Given the complexity of m6A regulation and the wide impact on gene expression, I do suggest some more experiments to be conducted and clarification in the paper before authors’ conclusions can be comfortably accepted for publication.

Thank you for appreciating the impact and novelty of the work.

1. Is the improved survival rate due to changes in m6A (the authors showed temporary increase of m6A in poly(A) after HS) ? Although *Mettl3*-siRNA clearly shows better survival rates, it is not clear whether this is due to m6A-dependent mechanisms. Given that m6A-independent role of *Mettl3* has been reported, another perturbation experiment targeting m6A is needed. Especially the authors show that *Ythdf*- and *Fmr1*-siRNA did not improve survival rate. Although the authors seem to have validated *Mettl14*-siRNA in supplementary figures, I could not find HS-related phenotypes in the knock-down animals (I may have overlooked). If the authors have not shown it, it will be important to include *Mettl14* knockdown effect in survival assay. Alternatively, targeting other methyltransferase complex to disrupt methylation activity can be used to serve the same purpose.

We appreciate your comments. We determine possible *Mettl3* independent roles in various ways. We confirmed our results using a catalytic mutant for *Mettl3* and observed similar increase in puromycin incorporation (Sup. Fig. 9). We also observed an increase in stress resilience compared to control with the *Mettl3* catalytic mutant (Fig. 2d), indicating that puromycin increase and heat stress resilience are dependent on the catalytic activity of *Mettl3* in the brain. We examined *Mettl14* knockdown and it also confers heat stress resilience and increased puromycin incorporation in brains compared to controls (Sup. Fig. 2 and 9).

We examined the nuclear reader protein *Ythdc1* and found that knockdown of *Ythdc1* increased puromycin incorporation levels and upregulation decreased puromycin levels (Fig. 6). Furthermore, *Ythdc1* knockdown increased stress resilience of animals, and increased Hsp70 protein and transcript levels in the brain (Fig. 7). As suggested by Reviewer 1, we have focused reader protein analysis on the *Ythdc1* protein.

2. The authors proposed *Mettl3*-mediated dampening effect may explain the brain-specific responses to stress. In Supplementary Figure S2a, the data showed that brain has “delayed” and “less dramatic” changes in heat shock chaperon proteins. This is very interesting, but how is the dynamics affected in *Mettl3*-siRNA animals? Only one time-point of 30min after HS was presented at the protein level. At RNA level, it will be interesting to extract the data for heat shock chaperon genes as an independent set. In addition, given the known uncoupling between transcription and translation of Hsp70, it will be necessary to know *Mettl3*-siRNA effect on the heat shock protein responses at protein levels.

We have now further analyzed the dynamics of the HS chaperone protein levels with *Mettl3* knockdown. At the protein level, we examined additional timepoints of 6 h and 24 h post HS for Hsp70, DnaJ-1, and stv chaperones. We find that *Mettl3* knockdown leads to a continued

increase in protein levels at these timepoints for HS chaperones (Fig. 5d). At the RNA level we find that *Mettl3* knockdown upregulated many heat shock chaperones at baseline (Hsp70, Hsp27, Hsp26, Hsp83, DnaJ-1). These HSP genes are also upregulated transcriptionally with *Ythdc1* knockdown (Fig. 7).

3. The differential expression results in the three groups (no-m6A, constant, and *Mettl3*-dependent m6A) are very interesting. However, the dynamic changes of methylation status at individual gene level was not sufficiently addressed. The individual IGV examples shown by the authors indicated loss of 5'UTR peaks and in the case of DnaJ-1, also loss in CDS (Fig 3 and 4), thus these individual examples can not explain the *Mettl3*-dependent increase m6A level shown in dot-blot in Fig 1. Some individual genes (possibly abundant ones) must have increased m6A modification to explain the results in Fig 1 and individual examples of those should be given.

Thank you for this critical comment. We conclude that the increase in methylation levels with heat stress seen by dot blot and LC/MS is due to an increase in the levels of *Mettl3*-dependent m⁶A methylated transcripts upon heat stress. We do not see significant increase in the peak levels on individual transcripts with heat shock. Overall there are only 6 significant individual methylation gene changes from the RADAR differential peak analysis comparing control to HS m⁶A (Supplementary Data 2). Interestingly, of the differential peaks, *stv* and *DnaJ-1* lose some individual methylation upon heat stress in their 5'UTR, yet they maintain a visible peak, unlike *Hsp68/70* that loses its 5'UTR peak entirely upon HS (Sup. Fig. 7).

We do see that the overall level of transcripts with *Mettl3*-dependent 5'UTR peaks increase with heat shock (47.9% of all *Mettl3*-dependent genes upregulated with HS, 10.7% downregulated), and this increase likely accounts for the increase we observe by LC/MS.

4. Puromycylation experiments were only conducted in basal state. Although global protein expression is known to be reduced during HS, which makes labeling efficiency lower thus detection harder, it is important to know the effect of *Mettl3*-siRNA during HS response. For example, 1.5 hrs HS + recovery condition can be used to allow sufficient labeling.

Thank you for this comment. We designed additional puromycin assays to assess the levels of puromycin incorporation with HS and *Mettl3* knockdown in the brain. We find that post-HS and 4 h of puromycin treatment that the levels of puromycin remain high in *Mettl3* RNAi samples. We observe that there is an increase in the levels of puromycin, consistent with previous *Drosophila* studies showing increased puromycin incorporation with HS¹⁶. Interestingly, we observe that baseline puromycin levels are increased in *Mettl3* RNAi brains to levels of control HS brains. These data are added to Fig. 6b.

Minor points:

1. Fig 2f. May the labels underneath the blot be UAS-*Mettl3*?

We apologize, this is a typo which has been corrected (Sup. Fig. 7).

2. Fig 4f. IGV view of *fl(2)d* show lots of reads in introns, possibly misalignment?

Thank you for this comment. We now show the full isoform list of *fl(2)d* and the mapping associated with each peak more clearly in Sup. Fig. 5.

3. Fig 4i. It seems that dtau protein WB data under hs 30 min condition is missing. We have now removed these data.

Reviewer #3 (Remarks to the Author):

In this manuscript, the authors investigate how m6A contributes to the brain's stress response in flies by using acute heat shock stress (HS). They find that m6A levels are elevated in the brain in flies and that they further increase in the adult brain after HS. Furthermore, they identify a subset of RNAs that are Mettl3 methyltransferase targets in the brain with 5'UTR m6A and show that a small number of these transcripts undergo dynamic m6A regulation after HS.

Furthermore, they show that Mettl3 depletion in flies leads to increased survival after HS. The authors use correlative analyses, global translation assays, and examination of a few specific m6A-containing RNAs to develop a model in which HS causes changes to m6A in some RNAs, and that loss of Mettl3 contributes to HS resilience by promoting the increased stability and increased translation of 5'UTR m6A targets. Although this is an interesting model, unfortunately the experiments do not support the broad conclusions in the manuscript. Many of the analyses remain correlative and are at times unclear or difficult to interpret. In my opinion, there are substantial experiments that would need to be done to support the authors' conclusions. I have discussed some of my concerns below and hope that the authors will find this helpful.

Thank you for appreciating the impact of our work and your constructive comments, bringing forward important considerations.

Major Comments

1. The authors only show a small number of RNAs that have Mettl3-dependent changes in m6A, which suggests that this is not a global m6A-dependent stress response but rather a handful of RNAs with differential methylation. I also have concerns about the Merip seq data; it does not seem to be rigorously analyzed (see below) and the vast majority of sites seem to be present with Mettl3 reduction. Are these peaks due to noise in the MeRIP analysis or to some new set of m6A sites/methylated RNAs for which m6A is deposited by a different MTase other than Mettl3? The authors state in the discussion that some MTase components direct m6A to distinct regions. While there is some evidence for this, in the end all Mtase components rely on Mettl3. Indeed, the mass spec data shows a very robust depletion of m6A in Mettl3 RNAi brains (Fig 1), suggesting that Mettl3 does in fact direct the methylation of most m6A sites and therefore leading to the assumption that this is due to noise in the MeRIP data.

Thank you for your comment. With the Reviewers' comments in mind, we have now focused the manuscript on the *Mettl3*-dependent m⁶A peaks and their response to heat stress and *Mettl3* knockdown. As noted in response to Reviewer 1, the field recognizes that it is extremely difficult to knockdown all m⁶A and the m⁶A field currently struggles with this observation of *Mettl3*-independent sites. Many reports and datasets indicate *Mettl3*-independent m⁶A sites persist even with CRISPR knockout of *METTL3*⁴⁻⁸. Recent independent work in *Drosophila* has also shown the presence of *Mettl3*-independent peaks^{1,2}; we observe *Mettl3*-independent m⁶A here with knockdown of *Mettl3*, and seen in Wang et. al.¹ and Kan et. al.² with *Mettl3*-null mutation. Due to the m⁶A persisting after *METTL3* knockout, in our and others studies⁴⁻⁸ it is speculated that other genes or isoforms⁹ may have a role in m⁶A deposition on mRNA^{7,10-13}. Given these issues, we have focused the manuscript on *Mettl3*-dependent m⁶A peaks and *Mettl3*'s role in stress. Given the concerns of the Reviewers and the issues, as noted, here we focused on *Mettl3*-dependent m⁶A peak genes vs all the other expressed genes (*Mettl3*-independent m⁶A and non-m⁶A genes).

2. It is unclear from the methods section how m6A peaks were called among replicates. Were only the peaks common to all biological replicates used? What is the variability in peaks

between each replicate? These steps are critical for ensuring the robustness of the MeRIP approach, as previous studies have shown that stochastic peak calling is a common feature of MeRIP datasets. Figure S3b shows high overlap between antibodies at the gene level, but overlap between individual replicates and at the peak level is not shown. Furthermore, I could not find a list of m⁶A peaks in their data files.

Thank you for this comment. The peaks common to all biological replicates were used. We used Metpeak and RADAR which take into account any statistically significant peak based on each biological replicate and the variability between replicates. We have included additional supplementary graphs to show the overlap of m⁶A enrichment sites between each replicate (Sup. Fig. 4h). We have also now included an additional dataset (Supplementary Data 2) with the m⁶A peak file outputs from RADAR.

3. It is unclear that the IP/input calculation properly accounts for read coverage in each MeRIP library. MeRIP is notorious for stochastic peak calling due to variability in IP efficiency. The gene tracks in figure 4f, for instance, show a lot of Mettl3-independent peaks and even for some the 5'utr mettl3-dependent peaks have a high amount of reads in Mettl3 depleted conditions. It would be more useful to show these gene tracks as separate tracks with IP and input reads on the same track so that we can see how much the IP is enriched relative to input. Also, how exactly was read depth taken into account when calculating IP/input? RADAR has methods for accounting for this but these were not mentioned in the Methods section. Also, the methods section says that a beta_cutoff (log fold change cutoff) of 0.5 was used, but then later it says that regions with a fold change <-1 were considered dependent peaks. It is not clear to me based on this description what exactly the thresholds were for fold-change determination.

Thank you for this comment. We now include additional figures with separated Input and m⁶A-IP tracks (Sup. Fig. 5c and 7c). When calculating IP enrichment over input, IGV tracks are CPM normalized. For heat map normalization, reads per million are normalized by the size of the bin, total reads, and library size. For RADAR peak calling the "Beta_cutoff" is the log fold change cutoff for selecting significant differential methylated bins, and default is +/- 0.5 log fold change for peak calling¹⁷. In the output file, RADAR will show the overall logFC of m⁶A/IP of an entire peak region. From this logFC output we filtered *Mettl3*-dependent m⁶A peaks that showed differential methylation levels of logFC <-1.

4. It is also not clear why the basal RRACH m⁶A consensus is not enriched in Merip peaks under basal conditions for either antibody. The authors claim that their motif analysis uncovered similar peaks to those found by Lence et al in S2R+ cells, but this does not seem to be the case under basal conditions. They do see RAC motifs in HS conditions, but curiously they have reported the 7th and 6th ranked motifs. Why? What were the top motifs?

Thank you for this comment. To further show our full *de novo* motif analysis we now include all the motifs in Supplementary Data 1. From *de novo* motif analysis, we see the same motif as found by Lence et. al.³ (our motif analysis was done using peaks combined from both HS and basal conditions due to peak locations similarity). We added additional motif analysis of *Mettl3*-dependent m⁶A and find similar motifs (Sup. Fig. 4e). We note that normally, for *Drosophila*, papers do not list all motifs, but only indicate that they see the m⁶A RAC motif. We could also do that, if the Reviewer prefers. Our data are fully consistent with previous work^{1-3,18}, just with more specific detail.

5. In figure 1a, the increase in m⁶A via dot blot seems to track with the amount of RNA loaded. In theory, the relative intensity should not change with RNA amount, but the quantification

indicates otherwise. Why is this? There are also wide variabilities in the intensity increase in m6A after HS across dot blots (eg, compare fig. 1a to 1c). I suggest including more replicates, especially since the MS data in panel 1b show a much less dramatic increase in m6A compared to the dot blots. In general, there is an over-reliance on dot blots throughout to confirm loss of m6A. Dot blots are a poor measure of m6A quantification and suffer from non-specificity issues related to recognition of non-m6A modifications by the antibodies. MS should be done, in particular to confirm the loss of m6A in *Mettl3* RNAi brain tissue, not merely whole heads.

Thank you for this comment. We have included additional replicates of the dot blots (Fig. 1a) and reduced the dot blots shown in the main figures and performed additional LC/MS experiments with additional HS recovery time points (Fig. 1b) and in the brain with *Mettl3* knockdown (Fig. 1c) that are in the main figures.

6. It is an overstatement to say that the brain has a dampened stress response based on the results of Figure S2, especially since the levels of Hsp70 are quite strongly induced in the brain after HS.

Thank you for this comment. What we mean specifically here is that the levels of Hsp70 upon heat stress are reduced in the brain relative to other tissues: whole heads and outer head capsules minus brain (Sup. Fig. 2c). The heat stress response of the brain is also delayed compared to the rapid onset of Hsp70 protein upregulation in head tissue and whole fly tissue (Sup Fig 2a). Third, the Hsp70 helper co-chaperones (HSP40/DNAJ-1 and Bag3/stv) levels do not increase at the protein level in the brain (Fig. 5d), unlike the increase seen in heads and whole fly tissue of these co-chaperones (Sup. Fig. 2d and 2e). These data indicate the brain does have a unique stress response compared to other *Drosophila* tissues. We have made this point more clear where it is made, in that the dampened stress is in comparison to other tissues like whole head and whole fly.

7. Figure 2e indicates a relatively modest effect of *Mettl3* RNAi on Hsp70 levels in the brain, and the western blot shown in the figure clearly has an increase in both tubulin and Hsp70. There is an over-reliance on western blot densitometry for quantification of protein amounts throughout the manuscript. This is concerning for some antibodies which have questionable specificity (eg, futsch, also dtau which appears to have two bands, one of which is cut off from the blot shown in Figure 4). A more robust readout for translation such as polysome analysis or quantitative western blot would help to confirm these results.

Thank you for this comment. We have included additional quantitative western blots for key Hsp70 western experiments (Sup. Fig. 7i) with serial dilutions of the brain samples. We also include a HS time course western with Hsp70, DnaJ-1, and stv (Fig. 5d and Sup. Fig. 7f-g). We hope the Reviewer finds these satisfying. Current techniques for polysome analysis in *Drosophila* require very large amounts of tissue (at least 1,000 *Drosophila* heads per replicate) for adequate analysis; it would be extremely challenging to obtain sufficient tissue of the brain, and for so many different samples.

8. The authors classify RNAs as no m6A, constant m6A, or *Mettl3* dependent m6A. It is unclear what the "constant" category is offering. Clearly the peaks in these RNAs are not *Mettl3*-dependent m6A peaks...so what do the authors think they are? To me it suggests very high background in their MeRIP-seq data (see comments above), which means that these constant RNAs are no different than non-methylated RNAs. Also, few genes show HS-dependent changes so it is unclear how the authors anticipate m6A altering so many transcripts when the methylation levels themselves do not change.

We thank the Reviewer for bringing this up and have significantly revised the manuscript in consideration of this point. Previous papers studying m⁶A in *Drosophila* have also seen peaks independent of various methyltransferase components¹⁹⁻²¹, so we note that these observations are consistent with previously published studies. However, as noted above in the response to Reviewer 1 (point 1), we have now focused the manuscript on the *Mettl3*-dependent m⁶A class of transcripts. Although the "constant" m⁶A peaks may represent deposition of m⁶A not mediated by *Mettl3*, as noted by the Reviewer, they may also represent background. For this reason, we included them in the non *Mettl3*-dependent gene set.

The Reviewer's point regarding the widespread transcription changes in stress despite a general lack of observed m⁶A differences is also well-taken. Overall, it is difficult to determine a change in methylation levels with stress, and many do not observe changes in methylation levels, as most m⁶A peaks remain constant between basal and heat shock conditions (Supplementary Data 2). Rather we observe changes in overall transcript levels of m⁶A marked transcripts upon stress. We sought to highlight that genes with *Mettl3*-dependent m⁶A peaks are elevated in expression level with the heat shock response compared to background non-*Mettl3* marked genes. We suggest that the increase in their transcript level (and therefore the overall increased amount of m⁶A present associated with the transcripts) is what is responsible for the rise of m⁶A during HS seen by dot blot and LC/MS.

9. It is not clear what the minimum fold change was for RNA-seq analysis in terms of identifying up/downregulated genes. Based on the mean log2FC listed in the text, as well as their gene expression plots for different classes of RNA, it appears that a lot of these transcripts are not drastically changed.

Thank you for this comment. We did not set a log fold change cut off for our RNA-seq analysis and only excluded genes with no significant fold change levels (i.e. Padj<0.05). While the mean log fold change levels are overall not drastic, the changes trend toward upregulation in levels. If we look more closely at specific transcripts such as HS chaperones *DnaJ-1* (logFC=5.02), *stv* (logFC=3.89), *hsp68* (logFC=9.96), these are examples of genes with *Mettl3*-dependent peaks with a large FC with HS. *Mettl3* knockdown upregulates the basal levels of these HS chaperones. This likely serves as a pre-conditioning to the stress response and may confer the stress resilience.

10. The authors show data that does not quite fit how they phrase the analysis. Line358 says, "more *Mettl3*-dependent m⁶A genes were differentially expressed both with HS and with *Mettl3* RNAi (18.2% of the DE genes in HS) compared to constant genes (6.5%) or no-m⁶A genes (6.3%)". These numbers are percentages of the HS vs Cntrl DE genes, but if you calculate the percentage instead relative to *Mettl3* RNAi vs Cntrl DE genes, you get much more similar numbers across all groups (50-60%).

Thank you for this comment. We have altered this figure and the language used in the manuscript to simplify and not overstate the data. We removed our comparisons of *Mettl3*-independent peaks and focus on *Mettl3*-dependent m⁶A (Fig. 4).

11. The data shown in Fig 5e,f (differential expression of genes in different GO/KEGG categories) includes for the most part a very small number of genes. Therefore these analyses should not be broadly interpreted to support more general claims of global effects of *Mettl3* and HS on RNA abundance and stress resilience.

Thank you for this comment. We were surprised that for neurogenesis terms of which 84 were significantly differentially expressed in *Mettl3* RNAi brains we see that 60 (71%) trend towards increased levels in *Mettl3* knockdown brains. Similarly for HS, we found that 257 neurogenesis genes with *Mettl3*-dependent peaks were differentially expressed with HS in the brain and 219 of them are increased with Heat stress. Globally, *Mettl3*-dependent genes account for 31.8% of the genes upregulated with heat stress (Sup. Fig. 7c). We included additional data to quantify the skew of genes differentially expressed upon heat stress or in *Mettl3* RNAi conditions. We find that compared to the background (all non *Mettl3*-dependent genes), the skew of increased expression for *Mettl3* dependent transcripts with heat stress and with *Mettl3* knockdown are statistically significantly (Fig. 4d-h).

12. The data linking m6A to Ythdf and Fmr1 are mostly correlative; while the authors show interesting effects of Ythdf and Fmr1 depletion on global protein synthesis, we do not know how many of these proteins are encoded by RNAs with m6A. This is important to know in order to fit the model that these proteins are “key regulators that promote *Mettl3*-dependent m6A inhibition of protein translation in the brain” as the authors conclude (lines 414-417). Also, since Fmr1 is known to inhibit translation through m6A-independent mechanisms, it is very likely that the effects they observe with Fmr1 depletion are occurring independently of the m6A pathway.

Given this and the other Reviewers’ comments, we removed these analyses and Fmr1 data from the manuscript to focus on reader protein Ythdc1 and its effect on heat stress.

13. The broad changes in protein synthesis after *Mettl3* depletion are inconsistent with the relatively few transcripts that show *Mettl3*-dependent m6A in the brain. Furthermore, the Sunset assays were not performed in HS brains, which would speak better to the potential role of *Mettl3* in regulating translational responses to stress.

Thank you for this feedback. We designed additional puromycin assays to assess the levels of puromycin incorporation with Heat stress and *Mettl3* knockdown in the brain. We find that post-HS and 4 h of puromycin treatment that the levels of puromycin remain high in *Mettl3* RNAi samples. We observe that there is an increase in the levels of puromycin consistent with previous *Drosophila* studies showing increased puromycin incorporation upon HS¹⁶. Interestingly we observe that *Mettl3* RNAi at baseline puromycin levels are increased to levels of control HS brains. These data are added to Fig. 6b.

Although *Mettl3*-dependent transcripts encompass ~14% of genes expressed in the brain, the total number of *Mettl3*-dependent genes is 2,120 genes, many of which are involved in dynamic signaling pathways and can have downstream effects on RNA turnover and protein translation. For comparison, the heat shock response, which is a robust stress response, upregulates 3,193 genes in the brain. 1,016 of those genes are *Mettl3*-dependent, so 31.8% of these genes are affected by *Mettl3*-dependent m⁶A, which is a significant number of transcripts.

14. The Sunset assays (Fig 6) have large variability in the control conditions across samples which is not reflected in the densitometry analysis. What is the source of this variability?

We apologize for the confusion. For the puromycin assay blots, the likely issue was that the densitometry analysis were shown side by side for blots that were run separately without adequate space in between. This is a visual error; we apologize and to correct this, we placed the densitometry analysis next to the blot that corresponds to it. Blots are run as alternating between control RNAi and experimental RNAi, or a separate blot for control vs upregulation. All blots are normalized to the corresponding biological replicate control on the blot; the exposure time of the blots is the source of variability between the control conditions. To fully observe the

decrease in puromycin levels in some conditions (such as upregulation of *Mettl3* brains), blots need to be exposed for longer time periods because the signal is so low. This is contrasted with conditions where there are increased changes in puromycin, and exposure times are shorter. We have tried to make this more transparent in the figure legends and methods.

15. The data showing the effect/lack of effect of *Ythdf1* and *Fmr1* on stress sensitivity should be shown. It would also be good to include the effect of *Mettl3* overexpression on stress resilience.

Thank you for this comment. In our original submission we included the effects of *Mettl3* upregulation (in restructured manuscript, this is now in Fig. 2c). In the restructured manuscript, we focus on nuclear reader *Ythdc1*. We find that *Ythdc1* knockdown shows similar stress resilience to *Mettl3* knockdown (Fig. 7).

Minor Comments

It is not clear to me how the heat shock was performed. The Methods sections describes a stress sensitivity assay with HS for 1.5h at 38.5C, but other experiments suggest 30min at 38.5C. The authors should specify when different conditions for the HS were used across experiments and why.

We have now added additional explanation in the methods to clarify. For heat shock survival, we follow the standards in the field. The standard is to use a longer severe heat shock of 1-2 h for survival of the flies post HS stress (see refs ^{14,15}). The standard to examine the molecular basis of the sensitivity to HS stress in *Drosophila* is 30 min. We consistently use a short heat shock as a mild stressor and a longer heat shock for a severe stressor to measure stress sensitivity

To further explain the reasoning behind these heat stress times and the method: For stress sensitivity, we HS flies in plastic vials in a water bath for 1.5 h and score the percentage surviving in vials of 20 flies 1 d post stress. The HS response is a remarkably fast response in *Drosophila* and protein chaperone changes are induced within minutes of the stress. There is a rapid upregulation of HS chaperones and helper chaperones to combat the stress, unlike mammalian cell systems which have a delayed response of ~6 h post mild HS stress. In flies, the 30 min heat shock is a mild heat shock that is sufficient to turn on HS response pathways and chaperones. We used the 30 min time point due to sufficient but non-lethal induction of critical heat shock protein chaperones.

The callouts to Figs 4f-l in the text are inaccurate.
We have fixed this typo.

Fig 6b: 5'UTR mislabeled as 3'UTR
This figure has been removed from updated MS.

It seems inaccurate to refer to head m6A peak RNAs as brain targets. I recognize that m6A is elevated in the brain relative to whole head/whole fly, but unless the authors show that head minus brain lacks m6A, or unless they perform MeRIP on brain specifically, they cannot know for sure whether these peaks come from the brain.

Thank you for this comment. We have edited the text (removed line 237 in original manuscript).

FINAL comments:

We thank the Editor and Reviewers for their careful reading and helpful comments and input on the manuscript. We hope with these extensive new data and modifications, the manuscript will now be appropriate for publication in Nature Communications.

References:

1. Wang, Y. *et al.* Role of Hakai in m⁶A modification pathway in Drosophila. *Nature Communications* **12**, 2159 (2021).
2. Kan, L. *et al.* A neural m⁶A/Ythdf pathway is required for learning and memory in Drosophila. *Nature Communications* **12**, 1458 (2021).
3. Lence, T. *et al.* m⁶A modulates neuronal functions and sex determination in Drosophila. *Nature* **540**, 242–247 (2016).
4. m⁶A mRNA methylation controls T cell homeostasis by targeting the IL-7/STAT5/SOCS pathways | Nature. <https://www.nature.com/articles/nature23450>.
5. Fu, Y. & Zhuang, X. m⁶A-binding YTHDF proteins promote stress granule formation. *Nature Chemical Biology* **16**, 955–963 (2020).
6. Dynamic landscape and evolution of m⁶A methylation in human | Nucleic Acids Research | Oxford Academic. <https://academic.oup.com/nar/article/48/11/6251/5837051?login=true>.
7. Schwartz, S. *et al.* Perturbation of m⁶A Writers Reveals Two Distinct Classes of mRNA Methylation at Internal and 5' Sites. *Cell Reports* **8**, 284–296 (2014).
8. Xiang, Y. *et al.* RNA m⁶A methylation regulates the ultraviolet-induced DNA damage response. *Nature* **543**, 573–576 (2017).
9. Poh, H. X., Mirza, A. H., Pickering, B. F. & Jaffrey, S. R. *Understanding the source of METTL3-independent m⁶A in mRNA*. 2021.12.15.472866
<https://www.biorxiv.org/content/10.1101/2021.12.15.472866v1> (2021)
doi:10.1101/2021.12.15.472866.
10. Lin, Z. *et al.* Mettl3-/Mettl14-mediated mRNA N⁶-methyladenosine modulates murine spermatogenesis. *Cell Res* **27**, 1216–1230 (2017).

11. Batista, P. J. *et al.* m6A RNA Modification Controls Cell Fate Transition in Mammalian Embryonic Stem Cells. *Cell Stem Cell* **15**, 707–719 (2014).
12. Tong, J. *et al.* Pooled CRISPR screening identifies m6A as a positive regulator of macrophage activation. *Science Advances* (2021) doi:10.1126/sciadv.abd4742.
13. Wei, G. *et al.* Acute depletion of METTL3 implicates N6-methyladenosine in alternative intron/exon inclusion in the nascent transcriptome. *Genome Res.* gr.271635.120 (2021) doi:10.1101/gr.271635.120.
14. Berson, A. *et al.* TDP-43 Promotes Neurodegeneration by Impairing Chromatin Remodeling. *Current Biology* **27**, 3579-3590.e6 (2017).
15. Multiple functions of Drosophila heat shock transcription factor in vivo. *The EMBO Journal* **16**, 2452–2462 (1997).
16. Deliu, L. P., Ghosh, A. & Grewal, S. S. Investigation of protein synthesis in Drosophila larvae using puromycin labelling. *Biology Open* **6**, 1229–1234 (2017).
17. Zhang, Z. *et al.* RADAR: differential analysis of MeRIP-seq data with a random effect model. *Genome Biol* **20**, 294 (2019).
18. Worpenberg, L. *et al.* Ythdf is a N6-methyladenosine reader that modulates Fmr1 target mRNA selection and restricts axonal growth in Drosophila. *The EMBO Journal* **40**, e104975 (2021).
19. Weng, Y.-L. *et al.* Epitranscriptomic m6A Regulation of Axon Regeneration in the Adult Mammalian Nervous System. *Neuron* **97**, 313-325.e6 (2018).
20. Engel, M. *et al.* The Role of m6A/m-RNA Methylation in Stress Response Regulation. *Neuron* **99**, 389-403.e9 (2018).
21. Ries, R. J. *et al.* m 6 A enhances the phase separation potential of mRNA. *Nature* **571**, 424–428 (2019).

REVIEWER COMMENTS

Reviewer #1 (Remarks to the Author):

The authors made a significant number of new experiments to improve the quality of their work and I acknowledge their effort. Nevertheless I still believe that the clarity of some findings should be improved.

1. Even though the authors removed the part of the Mettl3 independent m6A peaks the reading of the manuscript remains very confusing. At this stage there is no evidence to believe that these peaks are not the result of background signal. The fact that some signal is still present after Mettl3 KO by LCMS analysis likely reflects the presence of ribosomal RNA, which is impossible to fully remove from polyA preparation. The authors keep repeating Mettl3 dependent m6A peaks. I think they should make clear once what are the meaning of these peaks and then refer only to m6A peaks. Keeping Mettl3 dependent m6A peaks suggest that the other m6A peaks are relevant and can lead the field into a wrong direction. For instance in lane 245 it should be indicated "m6A transcripts are enriched in neuronal and signaling pathways". And so on.

In the same line, the GO term and KEGG pathway analyses should not be performed for non Mettl3 dependent modified brain genes. There is no reason to believe that these peaks are real and that the genes are different from other expressed, non modified genes.

Similarly, checking the expression of non Mettl3 modified gene (Hsf) is a nonsense. Just checking the expression of a non methylated transcript is sufficient.

Same comment for the RNA-seq analysis. The comparison should not be made with non-Mettl3 dependent m6A genes but with all other expressed genes.

2. After checking the nature of the Mettl3 catalytic mutant allele it appears that it is a strong deletion that removes a large part of the gene. I think this description is misleading and does not rule out a catalytic independent function. It was appropriate to use this allele to back up the RNAi experiment but this should not serve as an evidence to demonstrate that the catalytic activity is required. The fact that Mettl14 also has the same function already likely suggests that the effect goes through m6A.

3. Fig 4g: typo for misregulAted, and 281 genes down should be 220.

4. How many replicates were used for figure 5e?

5. In fig 6b the increased puromycin with the Mettl3 RNAi upon HS seems striking on the blot, which is poorly reflected in the quantification. Is it really a representative image?

6. Figs 2b and 7b. What is the difference between the daGAL4 control that were used that could explain the strong viability in fly survival (60% survival vs less than 20%)?

7. Fig 8. The scheme is confusing as Ythdc1 is within the Mettl3 RNAi rectangle. It should appear as a stand-alone part.

Reviewer #2 (Remarks to the Author):

In the revised version, the authors have added several sets of new experiments such as siRNAs against reader proteins, changes in chaperon protein expression under heat-shock conditions, Mettl3 catalytic mutants, and etc. The results consistently showed that the "m6A signaling-compromised" flies survived better after heat-shocks, possibly due to a pre-conditioned anti-stress effect by siMettl3, siMettl14, and siYthdc1. Furthermore,

by focusing on the more restricted set of Mettl3-dependent m6A peaks (which are mostly located to the 5'UTR), molecular responses such as the stress-response MAPK signaling pathway were identified associated with m6A regulation. The results from these new experiments have helped clarify the m6A-mediated response to HS tremendously.

All concerns raised in my previous review report have been well addressed except for one. The one concern that I could not agree with the authors' response to is concern #3. In my opinion, this is rather a critical concern for one of the main claims of this study: that HS triggers a temporary responsive increase in brain m6A level. My concern is that the increase in m6A level upon HS measured by dot-blot and mass spec was not supported by the MeRIP-seq data. Genes with significantly increased peaks after HS that can potentially account for the dot-blot and mass spec data were not identified (Fig 3). Although the authors explained the increase as "overall level of transcripts with Mettl3-dependent 5'UTR peaks increase with heat shock". To me, this result suggest rather the opposite that increased transcripts with less methylation will result in less m6A/A and dot blot intensity. My concern resonates with Reviewer 3's concern #5, that in the dot blot experiments, the changes in m6A level seems to "track with" the loaded RNA stained by methylene blue. The data panel in the revised Fig 1a remains so. Since the authors have been very careful with the purity of the extracted RNAs and included DNase twice to digest potential contamination of DNA, I suggest alternative explanations for the apparently conflicting results. For example, is it possible that the increased m6A sites went to unmappable (e.g. circRNA, sequence information lost during RT or amplification) portion?

Response to the Reviewer's concerns:

We thank the Editor and Reviewers for their constructive comments on our work. The remaining concerns of the Reviewers were the highlighting of *Mettl3*-independent m⁶A peaks, and lack of confirmation of m⁶A peaks with increased m⁶A density upon HS.

We have now fully addressed these concerns in a revised manuscript.

- We have revised the manuscript throughout to refer to “Mettl3-dependent peaks” as “m⁶A-peaks”. This included changes to the text and also figure panel titles.
- We have fixed typos and added any missing statistics or replicate explanations.
- Fig. 1: We have included clearer images of the methylene blue control blots (Fig. 1a,c).
- Fig. 2: We have included additional editing in the manuscript regarding the *Mettl3* “ΔCat” mutant (Fig.2d and associated text). We have edited text to say “ΔCat”, to clearly indicate deletion of catalytic domain. As the Reviewer notes, our data that the effect is dependent on methyltransferase complex includes effects with both *Mettl3* and *Mettl14*.
- Fig. 3: We performed additional analysis on m⁶A-RIP-seq data to show that m⁶A levels increase upon HS on *Mettl3*-dependent transcripts (Fig. 3a). We include additional scatterplots (Fig. 3b), as well as genome browser examples of genes that increase m⁶A upon HS in the brain (Supplementary Fig. 4d). To add statistical analysis to Fig. 3a,b we included violin plots with mean values and significance between conditions.
- Fig. 5: Fixed typos and renamed “Mettl3-dependent” to “m⁶A genes” vs “all other genes”
- Fig. 6: Reanalyzed raw blots of Fig. 6b and included clearer Ponceau S stained blots.
- Fig. 7: We include additional data of protein levels of *Ythdc1* upon HS and *Mettl3* knockdown (Fig. 7b), and overlap of m⁶A targets with genes differentially expressed with *Ythdc1* RNAi (Fig. 7e). We also repeated HS experiments for Fig. 7c to include *Mettl3* RNAi side by side with *Ythdc1* RNAi, to confirm the data with both knockdown conditions examined at the same time.
- Model: We separate out *Ythdc1* in its own box and clarify in the legend that *Mettl3* knockdown also leads to *Ythdc1* decreased levels.
- We have also made minor edits to the discussion to clarify and not overstate our findings.

To reiterate the significance of the work:

Here we reveal that m⁶A RNA modification stifles the brain's capacity to mitigate stress—greatly enhancing our understanding of m⁶A modification and stress response pathways essential to brain health, injury, and disease. The brain is uniquely vulnerable to cellular stress, and neurons often employ altered mechanisms to handle stress and combat degenerative disease to promote healthy brain aging. Current work in the field aims to define mechanisms to boost the brain's stress response. An emergent mechanism by which cells control downstream RNA stability and translation to proteins is the mRNA modification m⁶A. Particularly interesting is how m⁶A, and its methyltransferase *Mettl3*, shape the cellular stress response of the brain. Although a few *Drosophila* studies have been done on m⁶A, most studies focus on cultured epithelial S2 cells. In our data, we find that m⁶A targets are more highly expressed in the brain tissue compared to epithelial S2 cells, and suggests that regulation of m⁶A is best examined using brain tissue.

Specific novelty arising from our study includes:

1. *Mettl3* loss of function promotes heat stress survival, indicating m⁶A impedes the brain acute stress response.
2. M⁶A genes are a brain-enriched class that become upregulated upon heat stress and upon *Mettl3* knockdown.

3. *Mettl3* knockdown increases critical heat stress chaperone RNA and protein levels in the brain and increases protein translation.
4. A novel *ex vivo* brain actinomycin D assay shows that *Mettl3* knockdown influences RNA decay of critical heat stress chaperones.
5. *Mettl3* knockdown decreases levels of nuclear reader protein *Ythdc1*. *Ythdc1* knockdown allows for stress resilience, similar to *Mettl3* knockdown, and increases levels of heat stress chaperones.
6. This study is the first m⁶A-IP sequencing with heat stress *in vivo*, utilizing *Drosophila* heads with *Mettl3* manipulation, brain tissue RNA-sequencing with *Mettl3* knockdown and nuclear reader protein *Ythdc1* knockdown, with heat stress and stress recovery time-course. Thus in addition to the biological significance, this work provides several datasets of great value to the field.

Overall, this work has implications in a number of fields, as we present *Mettl3* and *Ythdc1* play important roles in regulating the brain's stress response.

Point by point Response:

We include the Reviewers comments in black and our response in blue.

Reviewer #1 (Remarks to the Author):

The authors made a significant number of new experiments to improve the quality of their work and I acknowledge their effort. Nevertheless I still believe that the clarity of some findings should be improved.

1. Even though the authors removed the part of the *Mettl3* independent m⁶A peaks the reading of the manuscript remains very confusing. At this stage there is no evidence to believe that these peaks are not the result of background signal. The fact that some signal is still present after *Mettl3* KO by LCMS analysis likely reflects the presence of ribosomal RNA, which is impossible to fully remove from polyA preparation. The authors keep repeating *Mettl3* dependent m⁶A peaks. I think they should make clear once what are the meaning of these peaks and then refer only to m⁶A peaks. Keeping *Mettl3* dependent m⁶A peaks suggest that the other m⁶A peaks are relevant and can lead the field into a wrong direction. For instance in lane 245 it should be indicated "m⁶A transcripts are enriched in neuronal and signaling pathways". And so on.

In the same line, the GO term and KEGG pathway analyses should not be performed for non *Mettl3* dependent modified brain genes. There is no reason to believe that these peaks are real and that the genes are different from other expressed, non modified genes.

Similarly, checking the expression of non *Mettl3* modified gene (*Hsf*) is a nonsense. Just checking the expression of a non methylated transcript is sufficient.

Same comment for the RNA-seq analysis. The comparison should not be made with non-*Mettl3* dependent m⁶A genes but with all other expressed genes.

Thank you for this comment. As requested by the Reviewer, to improve the clarity of the manuscript and the data, we have edited the figure panel titles and writing throughout the manuscript to remove "Mettl3-dependent m⁶A" and replaced simply with "m⁶A". Our analysis in the previous revision was between "m⁶A genes" and "all genes expressed but not marked by m⁶A"

(non-dependent)", for example labeling in the RNA-seq **Fig. 4**. We have changed the labeling in **Fig. 4** to improve clarity. Now the manuscript and figure read as "m⁶A" versus "All other genes". We include GO terms for "all other genes" in the supplement, to show the reader that when you do GO analysis on a group of brains genes (or background), you do not see the same neurogenesis or signaling GO terms.

2. After checking the nature of the *Mettl3* catalytic mutant allele it appears that it is a strong deletion that removes a large part of the gene. I think this description is misleading and does not rule out a catalytic independent function. It was appropriate to use this allele to back up the RNAi experiment but this should not serve as an evidence to demonstrate that the catalytic activity is required. The fact that *Mettl14* also has the same function already likely suggests that the effect goes through m⁶A.

Thank you for pointing this out. We have renamed "cat" in figures and manuscript to "ΔCat". We have edited the manuscript to make this point clear (**page 5**), and referred to the evidence on *Mettl14* to indicate the pathway is likely dependent on methyltransferase activity (**page 5**).

3. Fig 4g: typo for misregulAteD, and 281 genes down should be 220. We have fixed this typo.

4. How many replicates were used for figure 5e?

Fig. 5e is a representative time course graphs taken from one biological replicate; the 4 hr timepoints from all 6 biological replicates of the experiments are in **Fig. 5f**. Plotted in **Fig. 5e** are three technical replicates with mean and SD from the technical replicates. **Fig. 5f** shows data collected from 6 individual biological replicates. We have indicated this correctly in the figure legends. Thank you for pointing this out.

5. In fig 6b the increased puromycin with the *Mettl3* RNAi upon HS seems striking on the blot, which is poorly reflected in the quantification. Is it really a representative image?

Thank you for noting this. We have now carefully re-analyzed our raw blots and found that the levels of *Mettl3* RNAi puromycin post HS should be much higher by quantification. We include new statistics and a clearer image of the Ponceau S stain used as a loading control.

6. Figs 2b and 7b. What is the difference between the daGAL4 control that were used that could explain the strong viability in fly survival (60% survival vs less than 20%)?

Thank you for noting this. Fly survival after heat shock will depend on the precise temperature of the water bath used and the precise temperature of the incubator where the animals are kept, and who precisely does the experiment. Those experiments also happened to have been performed at very different times (at least a year apart). To prevent confusion, we now have repeated the experiment with all genotypes side by side of *Ythdc1* RNAi and *Mettl3* RNAi and controls. We include these data as a replacement to the data in **Fig. 7b**, as that control in that specific experiment seemed outside the normal range of susceptible to HS (although the data were clearly highly reproducible in regards to HS resilience within that experiment).

7. Fig 8. The scheme is confusing as *Ythdc1* is within the *Mettl3* RNAi rectangle. It should appear as a stand-alone part.

Thank you for this point. We have edited the model figure to highlight *Ythdc1* as a stand-alone part of the model, in the figure and in the legend. We now also include additional evidence that

Mettl3 knockdown decreases *Ythdc1* levels by including new western immunoblot data that show that the protein levels are also affected (**Fig. 7b**).

Reviewer #2 (Remarks to the Author):

In the revised version, the authors have added several sets of new experiments such as siRNAs against reader proteins, changes in chaperon protein expression under heat-shock conditions, *Mettl3* catalytic mutants, and etc. The results consistently showed that the “m6A signaling-compromised” flies survived better after heat-shocks, possibly due to a pre-conditioned anti-stress effect by si*Mettl3*, si*Mettl14*, and si*Ythdc1*. Furthermore, by focusing on the more restricted set of *Mettl3*-dependent m6A peaks (which are mostly located to the 5'UTR), molecular responses such as the stress-response MAPK signaling pathway were identified associated with m6A regulation. The results from these new experiments have helped clarify the m6A-mediated response to HS tremendously.

All concerns raised in my previous review report have been well addressed except for one. The one concern that I could not agree with the authors' response to is concern #3. In my opinion, this is rather a critical concern for one of the main claims of this study: that HS triggers a temporary responsive increase in brain m6A level. My concern is that the increase in m6A level upon HS measured by dot-blot and mass spec was not supported by the MeRIP-seq data. Genes with significantly increased peaks after HS that can potentially account for the dot-blot and mass spec data were not identified (Fig 3). Although the authors explained the increase as “overall level of transcripts with *Mettl3*-dependent 5'UTR peaks increase with heat shock”. To me, this result suggest rather the opposite that increased transcripts with less methylation will result in less m6A/A and dot blot intensity. My concern resonates with Reviewer 3's concern #5, that in the dot blot experiments, the changes in m6A level seems to “track with” the loaded RNA stained by methylene blue. The data panel in the revised Fig 1a remains so. Since the authors have been very careful with the purity of the extracted RNAs and included DNase twice to digest potential contamination of DNA, I suggest alternative explanations for the apparently conflicting results. For example, is it possible that the increased m6A sites went to unmappable (e.g. circRNA, sequence information lost during RT or amplification) portion?

Thank you for this comment. We have gone back and re-analyzed our m⁶A-IP sequencing data to include only m⁶A sites that are dependent on *Mettl3*-m6A, as requested by Reviewer 1. Our manuscript now simplifies this to “m⁶A” genes. Through this analysis we see that m⁶A levels (normalized to input control), show a slight increase compared to basal m6A levels (**Fig. 3a**). Note that the extent of increase is consistent with m⁶A mammalian cell culture HS data¹. We also performed additional analysis of m⁶A -IP read coverage (normalized to input) in the 5'UTR of m⁶A genes in HS versus basal conditions. Overall, we see a slight increase in m⁶A levels with HS (**Fig. 3a-b**); however, when using the differential peak analysis program RADAR, many of these peaks are below the peak threshold level (by contrast, a strong knockdown or change in m⁶A levels such as with *Mettl3* knockdown is easily detected using programs like RADAR (**Supplementary Data 2**)). We have now included a scatterplot of the genes; there is a clear shift towards increased m6A with HS (**Fig. 3b**). We also include a violin plot that indicates for each m⁶A gene the normalized 5'UTR coverage in the 5'UTR (**Fig. 3b**). This plot indicates the mean m⁶A level and that there is a significant increase in the overall m⁶A levels in HS vs control conditions. Additionally, we have now included examples of MAPK genes illustrating the increased m⁶A peaks in the 5'UTR with HS (**Supplementary Fig. 4d**).

Taken together, these data support an increase in m⁶A levels upon HS. Again, our data show this by dot blot (which over-emphasizes the increase) (**Fig. 1a**), by mass spec (**Fig. 1b-c**, **Supplementary Fig. 1a**), and by analysis of the sequencing data (**Fig. 3a-b**, **Fig. 4a**, **Supplementary Fig. 4d**). We thank the Reviewer for pointing this out so that we have had the opportunity to revisit the data and provide more compelling examples and analyses. Although m⁶A shows only a slight increase with HS in the brain, we focus on the more robust effect of *Mettl3* knockdown on transcripts critical to the stress response.

FINAL COMMENTS:

We thank the Reviewers and the Editor for their comments and suggestions, which we feel have greatly improved the manuscript. We hope with these revisions, the Editor and Reviewers will find the manuscript acceptable for publication in ***Nature Communications***.

References:

1. Zhou, J. *et al.* Dynamic m⁶A mRNA methylation directs translational control of heat shock response. *Nature* **526**, 591–594 (2015).

REVIEWERS' COMMENTS

Reviewer #1 (Remarks to the Author):

The authors fully answered my remaining concerns and I congratulate them for the interesting findings.

Reviewer #2 (Remarks to the Author):

The authors have addressed all my concerns in the revised manuscript. I support the publication of this study.

Response to the Reviewer's:

Thank you to the reviewers for support of our work and for the constructive comments and suggestions that have greatly improved the quality of this study.

Reviewers' CommentsReviewer #1 (Remarks to the Author):

The authors fully answered my remaining concerns and I congratulate them for the interesting findings.

Reviewer #2 (Remarks to the Author):

The authors have addressed all my concerns in the revised manuscript. I support the publication of this study.